# Testosterone modulates multispectral oscillatory activity serving performance of motor sequences in typically developing youth

Jackson Derby[1,2], Thomas W. Ward[1,2,3] , Nathan M. Petro[1,2], Jake J. Son[1,2,4] , Grace C. Ende[1,2], Danielle L. Rice[1,2], Anna T. Coutant[1,2], Erica L. Steiner[1,2], Vince D. Calhoun[5], Yu-Ping Wang[6], Julia M. Stephen[7], Elizabeth Heinrichs-Graham[1,2,3] and Tony W. Wilson[1,2,3,4]

[1] *Institute for Human Neuroscience, Boys Town National Research Hospital, Boys Town, NE, USA*
[2] *Center for Pediatric Brain Health, Boys Town National Research Hospital, Boys Town, NE, USA*
[3] *Department of Pharmacology and Neuroscience, Creighton University School of Medicine, Omaha, NE, USA*
[4] *College of Medicine, University of Nebraska Medical Center, Omaha, NE, USA*
[5] *Tri-Institutional Center for Translational Research in Neuroimaging and Data Science (TReNDS), Georgia State University, Georgia Institute of Technology, and Emory University, Atlanta, GA, USA*
[6] *Department of Biomedical Engineering, Tulane University, New Orleans, LA, USA*
[7] *Mind Research Network, Albuquerque, NM, USA*

Handling Editors: Richard Carson & James Coxon

The peer review history is available in the Supporting Information section of this article (https://doi.org/10.1113/JP289314#support-information-section).

**Abstract figure legend** Increases in testosterone are associated with developmental changes in motor-related neural oscillations in youth.

J. Derby and T. W. Ward contributed equally.

The Journal of Physiology

**Abstract** Motor control is critical to daily functioning and undergoes significant refinement throughout youth. Although the effects of age on the underlying brain circuitry have been well characterized, fewer studies have examined the influence of pubertal factors on this maturation. This distinction is crucial, as an increase in hormones like testosterone may be a better predictor of neural development than age alone and thus more informative in understanding the functional changes observed during youth. To this end, we enrolled 69 typically developing participants (10–17 years old) who performed a motor sequencing task during magnetoencephalography (MEG) and provided saliva samples for testosterone assays. The neural oscillations serving motor control were imaged and examined using whole-brain ANCOVAs with sex and testosterone levels as factors of interest, controlling for age. Our key findings indicated sex-specific effects of testosterone, such that increasing testosterone levels in males, but not females, were associated with weaker beta oscillations in the prefrontal cortex. Increasing testosterone levels were also associated with weaker alpha oscillations across boys and girls in several frontal and inferior parietal regions. Additionally, there were sex-specific effects of testosterone on motor-related gamma oscillations in cerebellar regions, such that increasing testosterone was correlated with stronger activity in males only. Finally, neural responses in several of these regions were significantly coupled with reaction time. These findings suggest that testosterone has important effects on motor-related neurophysiology above and beyond age, and that these changes may serve the functional refinement of motor sequencing during this developmental period.

(Received 22 May 2025; accepted after revision 24 February 2026; first published online 16 March 2026)

**Corresponding author** T. W. Wilson: Brookhouser Endowed Chair & Director, Institute for Human Neuroscience, Boys Town National Research Hospital. Email: tony.wilson@boystown.org

## Key points

- The transition from childhood to adolescence is a critical period for brain development and is characterized by improvements in motor performance and increases in levels of pubertal hormone such as testosterone.
- Previous studies have shown that movement-related neural oscillations in extended motor regions undergo functional refinement during this period, but whether these changes are coupled to developmental increases in testosterone remains unclear.
- Herein, magnetoencephalography was used to derive whole-brain functional maps of movement-related oscillations during a motor sequencing task in 69 typically-developing youth, who also provided a sample for assaying testosterone.
- Testosterone levels, controlling for age, were coupled in a sex-specific way to neural oscillatory responses in higher-order brain regions serving motor planning and execution, with activity in several of these regions being correlated with behavioural performance.
- These findings suggest that testosterone plays a critical role in the maturation of higher-order motor circuitry during the pubertal transition period.

**Jesse Derby** completed a biomedical engineering degree at Cornell University and a two-year research assistantship at the Institute for Human Neuroscience. Current research interests include the effects of early childhood trauma on the structural and functional development of neural circuitry and brain dynamics, and other interests include philosophy, particularly studies in ethics and metaphysics, as well as recreational biking and reading. **Thomas Ward** completed his master's in pharmacology and neuroscience at Creighton University while conducting his research at the Institute for Human Neuroscience. His research focuses on brain dynamics and the use magnetoencephalographic (MEG) measures to study the development of executive function and motor control, with the ultimate goal of developing neuroscience-informed physical therapy approaches.

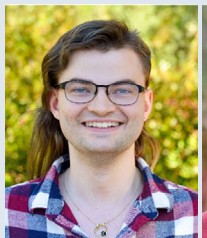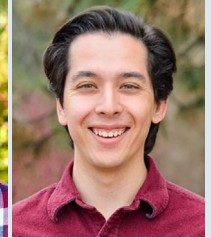

## Introduction

Motor control is essential to nearly every facet of human life, emerging early in development and gradually improving into adulthood. This acquisition of proficiency enables complex interactions with the outside world (Gentilucci & Volta, 2008; Grillner & El Manira, 2020; Knill, 2005), and contributes to cognitive development in youth (Bornstein et al., 2013; Koziol & Lutz, 2013; Schulte et al., 2020). In terms of the underlying neurophysiology, the planning and execution of voluntary actions are supported by neural population-level oscillations in the alpha ($\sim$8–15 Hz), beta ($\sim$15–30 Hz) and gamma bands ($\sim$30–90 Hz; (Heinrichs-Graham & Wilson, 2015; Pfurtscheller & Lopes da Silva, 1999; Wilson et al., 2010; Wilson et al., 2011). Specifically, several hundred milliseconds before movement onset, a robust decrease in beta power relative to baseline, termed an event-related desynchronization (ERD), emerges in the primary motor cortices and persists throughout the duration of the movement (Arif et al., 2024, 2026; Cheyne, 2013; Heinrichs-Graham & Wilson, 2015; Wilson et al., 2010). This so-called peri-movement beta ERD has long been associated with both motor planning and execution, and is known to be modulated by the certainty of the motor plan and other factors (Grent-'t-Jong et al., 2014; Heinrichs-Graham & Wilson, 2015; Heinrichs-Graham et al., 2016, 2020; Tzagarakis et al., 2010, 2015). A similarly extensive peri-movement alpha ERD also emerges, typically occurring slightly posterior to the beta ERD (Heinrichs-Graham & Wilson, 2015; Jurkiewicz et al., 2006; Pfurtscheller & Lopes da Silva, 1999; Salmelin et al., 1995). In contrast to beta, the alpha ERD is thought to reflect sensory feedback and sensorimotor integration (Babiloni et al., 2008, 2014; Erla et al., 2012; Silva et al., 2012; Van Diepen et al., 2016). In contrast to these temporally extended ERDs, a transient increase in gamma power, which is generally referred to as the movement-related gamma synchronization (MRGS), emerges in the contralateral motor cortex (Cheyne et al., 2008; Huang et al., 2025; Kurz et al., 2014, 2024; Meehan et al., 2023; Son et al., 2025; Webert et al., 2025). The MRGS is tightly locked to movement onset and was initially thought to serve basic movement execution (Cheyne & Ferrari, 2013; Muthukumaraswamy, 2010), but later work extended its role to higher-order cognitive control and attentional processes (Gaetz et al., 2013; Grent-'t-Jong et al., 2013; Heinrichs-Graham et al., 2018; Son, Arif, et al., 2024; Spooner & Wilson, 2022; Wiesman et al., 2020, 2021). Finally, after movement termination, there is a sharp increase in beta power referred to as the post-movement beta rebound (PMBR; Fry et al., 2016; Gaetz et al., 2010; Heinrichs-Graham et al., 2017; Jurkiewicz et al., 2006; Wilson et al., 2011), which is thought to reflect sensory feedback following the completed movement.

Evidence from several studies suggests that the oscillatory dynamics underlying motor control (Gaetz et al., 2010; Heinrichs-Graham et al., 2020; Heinrichs-Graham et al., 2018; Killanin et al., 2023; Trevarrow et al., 2019), as well as other cognitive processes (Hwang et al., 2016; Son, Killanin, et al., 2024; Taylor et al., 2021), scale with development. Most of these studies used chronological age as a marker of development, though there is compelling evidence suggesting other factors, such as the increase in circulating levels of pubertal hormones like testosterone and dehydroepiandrosterone (DHEA), may contribute to the functional refinement of oscillatory activity in adolescence. Though not direct measures of pubertal stage, these hormones are associated with pubertal development and correspond well with physical measures of puberty (Huang et al., 2012; Omary et al., 2025; Shirtcliff et al., 2009; Tyborowska et al., 2016). Further, pubertal onset and sharp increases in the level of these hormones is thought to trigger a critical period of brain development (Blakemore et al., 2010; Goddings et al., 2014; Larsen & Luna, 2018; Romeo, 2003; Sisk & Zehr, 2005), during which key structural and functional changes in the association cortices facilitate improvements in cognitive abilities (Fung et al., 2020; Fung et al., 2022; Killanin et al., 2024; Penhale et al., 2022; Picci et al., 2023; Schulte et al., 2020; Vijayakumar et al., 2021). Despite this, only a few studies have examined how these pubertal factors impact motor-related oscillations (Fung et al., 2022; Killanin et al., 2023).

To date, studies linking neural oscillations and pubertal hormones during development have typically used magnetoencephalography (MEG). Some of the earliest were focused on visual attention and found that increasing testosterone levels were associated with maturation of oscillatory responses serving visuospatial processing (Fung et al., 2020). A later study showed that testosterone was not related to the MRGS or PMBR responses during an isolated single-finger tapping task, but did directly predict beta ERD magnitude in the primary motor cortex, and mediated the relationship between age and ERD (Fung et al., 2022). These findings were confirmed and extended in a later study of movement sequences, which found that testosterone mediated the effect of age on both the beta ERD and behavioural performance (Killanin et al., 2023). These studies clearly support the view that testosterone levels are tightly coupled to developmental improvements in cognitive and motor function, as well as the underlying oscillatory activity. However, other aspects of this relationship remain less clear due to limitations in past work. For example, Killanin et al. (2023) focused on beta responses in the primary motor cortex and did not examine other motor-related oscillatory responses or brain regions outside the primary motor cortex. Likewise, the motor-related work of Fung and colleagues focused on only the primary response during a basic task and

did not consider other brain regions (Fung et al., 2022). Thus, whether testosterone impacts the refinement of motor-related oscillatory activity in secondary motor and other higher-order regions, its role in developmental sex differences, and possible involvement in motor sequence-related gamma responses remains largely unstudied due to previous work being highly focused on the primary motor cortex and/or the beta ERD response.

In the present study, we integrate measures of endogenous testosterone based on salivary hormone assays and high-density MEG recordings of typically developing youth performing a motor sequencing task. While our sample overlaps considerably with Killanin et al. (2023), we have conducted a full reanalysis of these data to address outstanding questions. Specifically, whereas Killanin et al. (2023) focused on the beta ERD in the primary motor cortex, the current study leverages a data driven approach to derive significant motor-related oscillatory responses and thus is sensitive to the alpha and beta ERD responses, the PMBR and the MRGS discussed above. Unlike prior work in this area, we also take a whole-brain, voxel-wise approach that is sensitive to both testosterone and sex difference effects across the brain. Thus, while our sample overlaps, our analysis approach almost completely differed and this enabled us to address novel aims, including identification of the independent and interactive effects of testosterone and sex on motor behaviour and the neural oscillatory dynamics underlying motor control at the whole-brain level. Above and beyond the effects of age, we hypothesized that testosterone levels would be significantly coupled to the strength of beta and gamma oscillations in brain regions associated with top-down executive control and that these effects would differ by sex. Further, we hypothesized that the strength of these neural responses would be related to behavioural performance on the task.

## Methods

### Ethical approval

All experimental procedures conformed to the standards set by the latest version of the *Declaration of Helsinki*, except for registration in a database. The study protocol (no. 503-15-EP) was approved by the Institutional Review Board (IRB) of the University of Nebraska Medical Center. Written informed consent was obtained from the child's parent or legal guardian and assent was obtained from each child following a complete description of the study.

### Participants

A total of 69 typically developing youth (36 female) between 9 and 17 years of age (mean age: 13.06; SD: 1.66) were recruited through the Omaha (NE, USA) enrollment site of the Developmental Chronnecto-Genomics (DevCoG) study. Potential participants were excluded from the study based on the following criteria: any diagnosed neurological or psychiatric disorder, any medical illness associated with CNS dysfunction, history of head trauma, current substance use, pregnancy and standard MEG/MRI contraindications (e.g., metallic implants that could interfere with or be a MEG or MRI safety concern). Of note, all participants in the current study were also included in the larger studies of Fung and colleagues, which used different cognitive tasks and analysis pipelines (Fung et al., 2021, 2022). Likewise, 58 participants in the current study were part of a recent study that examined the same data (Killanin et al., 2023). Importantly, data in the current study were completely reanalysed from the raw data onward using a different analysis pipeline (see Introduction).

### Salivary testosterone collection and measurement

At least 2.0 mL of unstimulated saliva was collected from each participant. Participants were instructed to refrain from consuming any food, liquids or chewing gum for at least an hour before providing the saliva sample, and generally completed the study in the afternoon (mean = 15:45, SD = 3.23 h). Specifically, participants passively drooled into an Oragene DISCOVER (OGR-500; DNA Genotek, Ottawa, Ontario, Canada) collection tube until liquid saliva exceeded the fill line indicated on the tube. A single-channel pipette was then used to extract 0.5 mL from the collection tube (prior to the release of the protease inhibitors for long-term storage), and this 0.5 mL sample was immediately transferred into a labelled micro-centrifuge tube and placed in a $-20°C$ freezer for storage. All samples were assayed in duplicate using a commercially available enzyme immunoassay kit for salivary testosterone (Salimetrics, Carlsbad, CA, USA) at the University of Nebraska-Lincoln Salivary Biosciences Laboratory. The assay kit had a sensitivity of 1 pg/mL, with a range of 6.1–600 pg/mL; the intra - and inter-assay coefficients of variation were 5.28% and 8.93%, respectively. The average of the duplicate tests was used for analysis in the present study. Testosterone levels across our sample were not normally distributed, and thus these data were transformed to their base-10 logarithmic values ($\log_{10}$) prior to analysis.

### Experimental paradigm

We used a motor sequencing paradigm designed by Heinrichs-Graham and Wilson (2015) during MEG recording. This task has been extensively used in both adults and children (Heinrichs-Graham & Wilson, 2016;

Heinrichs-Graham & Wilson, 2015; Heinrichs-Graham et al., 2020; Heinrichs-Graham et al., 2018; Ward et al., 2023). Participants were instructed to place their right hand on a button pad and view a fixation cross for 3750 ms during MEG recording. Three numbers, each corresponding to a digit on the hand (i.e., '1' for index, '2' for middle, etc.), were shown on the screen for 500 ms. The numbers on the screen then changed colour, indicating the cue to move, and participants had 2250 ms to tap out the sequence with the corresponding fingers as quickly and accurately as possible. After the 2250 ms, the numbers disappeared, leaving only the fixation cross on the screen. Following the 3750 ms fixation period, the next three numbers appeared (Fig. 1). There was a total of 160 trials, resulting in approximately 16 min of recording time.

## MEG and MRI data acquisition

Functional MEG data were collected during performance of the motor sequencing task using a 306-sensor MEGIN MEG system (Helsinki, Finland) equipped with 204 planar gradiometers and 102 magnetometers. Data were acquired at a 1 kHz sampling rate with an acquisition bandwidth of 0.1–330 Hz in a one-layer magnetically shielded room with active shielding engaged. Prior to recording, four coils were attached to the participant's head and localized along with fiducial and scalp surface points using a 3D digitizer (FASTRAK, Polhemus Navigator Sciences, Colchester, VT, USA). Once the participants were positioned for MEG recording, an electric current with a unique frequency label (i.e., 322 Hz) was fed to each coil, inducing a measurable magnetic field. This allowed each coil, and therefore the head, to be localized in reference to the MEG sensor array throughout the entire recording session.

A 3.0 T Siemens Prisma scanner (Erlangen, Germany) equipped with a 32-channel head coil was used to collect high-resolution structural T1-weighted MRI data from each participant (repetition time (TR): 24.0 ms; time to echo (TE): 1.96 ms: field of view: 256 mm; slice thickness: 1 mm with no gap; in-plane resolution: $1.0 \times 1.0$ mm). Structural volumes were aligned parallel to the anterior and posterior commissures and underwent inhomogeneity correction, segmentation, surface reconstruction and transformation into standardized space. Each participant's MEG data were co-registered with their MRI data using the Brain Electrical Source Analysis (BESA) MRI software (v3.0; Gräfelfing, Germany). After source imaging, each subject's functional images were also transformed into standardized space using the transform previously applied to the structural MRI volume, and spatially resampled.

## Time–frequency transformation and statistical analysis

MEG data were subjected to environmental noise reduction and corrected for head motion using the signal space separation method with a temporal extension (Taulu & Simola, 2006). Eye blinks and cardiac artifacts were removed from the data using signal space projection, which was accounted for during source reconstruction (Uusitalo & Ilmoniemi, 1997). The continuous magnetic time series was then divided into 6400 ms epochs, including a baseline window from −2250 to −1750 ms, with the onset of movement defined as 0 ms. Epochs containing artifacts were then rejected based on a fixed amplitude and gradient threshold method that was set per participant and supplemented with visual inspection. Briefly, in MEG, the raw signal amplitude is strongly affected by the distance between the brain and the MEG sensor array, as the magnetic field strength falls off sharply as the distance from the current source (i.e., brain) increases. Thus, to account for this and other sources of

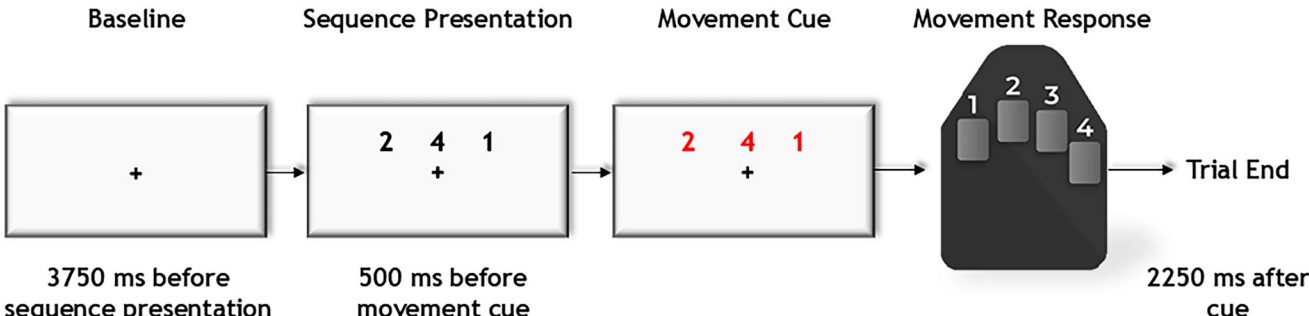

**Figure 1. Motor sequencing task paradigm**
Participants were instructed to place their right hand on a button pad and fixate on a centrally presented crosshair for 3750 ms. After this period, a sequence of three numbers, each corresponding to a digit on the hand (i.e., '1' for index, '2' for middle, etc.), was displayed for 500 ms. The numbers then changed colour, indicating the cue to move, with the sequence disappearing after 2250 ms (i.e., the participant had 2250 ms to complete the sequence, but was instructed to respond as quickly and accurately as possible).

variance across participants, we used individualized thresholds based on the signal distribution for both amplitude (mea*n* = 1562.5 fT/cm; SD = 447.11) and gradient ($M$ = 614.32 fT/(cm ms); SD = 232.02) to reject artifacts. On average, 121.80 (SD = 22.21) trials remained after artifact rejection. The number of accepted trials did not differ by sex and was not correlated with age, but was positively correlated with log-transformed testosterone ($r_{66}$ = 0.30, $P$ = 0.013). Since the number of trials can affect the signal-to-noise ratio (SNR) and thus bias effects of interest, we covaried out the square root of the number of trials per person in all functional mapping analyses to mitigate the effect of possible SNR differences (Pulliam et al., 2024).

Artifact-free epochs were transformed into the time–frequency domain using complex demodulation (Kovach & Gander, 2016; Papp & Ktonas, 1977). We chose to use complex demodulation in this study because it is computationally efficient and offers key advantages over other methods, including low spectral leakage and better transient detection (Kovach & Gander, 2016). Briefly, complex demodulation works by first transforming the signal into the frequency space using a fast Fourier transform. This results in a frequency spectrum, inherently containing the same power and cross spectrum information as the original signal. This frequency spectrum is then (de)modulated in a step-wise manner to adopt the centre frequency of a series of complex sinusoids with increasing carrier frequencies, in a process termed 'heterodyning'. These resulting signals are then low-pass filtered to reduce spectral leakage, and thus the nature of this filter inherently determines the time and frequency resolution of the resulting data. For this study, the time–frequency analysis was performed with a frequency-step of 2 Hz and a time-step of 25 ms between 4 and 90 Hz, using a 4 Hz lowpass finite impulse response filter. The resulting spectral power estimations per sensor were averaged across trials, generating time–frequency plots of mean spectral density. The sensor-level data were then normalized per time–frequency bin using the respective bin's baseline power, which was calculated by averaging the power during the 500 ms baseline period. The specific time–frequency bins used for source reconstruction were determined using a mass univariate approach based on the general linear model. To reduce the risk of false-positive results while maintaining reasonable sensitivity, a two-stage procedure was followed to control for Type-1 error. In the first stage, two-tailed paired-sample Student's $t$ tests against baseline were conducted on each data point, and the output spectrogram of $t$-values was thresholded at $P < 0.05$ to define time–frequency bins containing potentially significant oscillatory deviations across all participants. In stage 2, time–frequency bins that survived the threshold were clustered with temporally and/or spectrally neighbouring bins that were also above the threshold ($P < 0.05$), and a cluster value was derived by summing the $t$-values of all data points in the cluster. Nonparametric permutation testing was then used to derive a distribution of cluster values, and the significance level of the observed clusters (from stage one) was tested directly using this distribution (Ernst, 2004; Maris & Oostenveld, 2007). This testing was conducted from 4 to 90 Hz across all participants to assess for theta, alpha, beta and high gamma frequency bands, which have been previously linked to motor processes (Fung et al., 2022; Spooner et al., 2022; Wilson et al., 2010). For each comparison, 10,000 permutations were computed. Based on these analyses, the time–frequency windows containing significant oscillatory events relative to baseline across all participants were beamformed. For further details on our data processing pipeline, see Wiesman & Wilson (2020).

## MEG source imaging

Oscillatory neural responses were imaged using the dynamic imaging of coherent sources (DICS) beamformer (Gross et al., 2001; Van Veen et al., 1997), which utilizes spatial filters in the time–frequency domain to calculate voxel-wise source power for the entire brain volume. The single images were derived from the cross-spectral densities of all combinations of MEG gradiometers averaged over the time–frequency range of interest and the solution of the forward problem for each location on a grid specified by input voxel space. Following convention, we computed noise-normalized source power for each voxel per participant using active (i.e., task) and passive (i.e., baseline) periods of equal duration and bandwidth (Hillebrand et al., 2005) at a resolution of $4.0 \times 4.0 \times 4.0$ mm. Such images are referred to as pseudo-$t$ maps, with units (pseudo-$t$) reflecting noise-normalized power differences (i.e., active *versus* baseline) per voxel. Individual participant maps were examined and outliers (e.g., those exhibiting non-physiological patterns of activity) were removed. To assess the neuroanatomical basis of the significant oscillatory responses identified through the sensor-level analysis, grand-average whole-brain pseudo-$t$ maps were computed, excluding outliers. MEG preprocessing and source imaging used the BESA software (Research v7.1, Statistics v2.1, MRI v3.0; Gräfelfing, Germany).

## Statistical analyses

Behavioural performance was analysed using $1 \times 2$ ANCOVAs on each behavioural metric (i.e., accuracy, reaction time, movement duration) separately, with sex as a between-subjects factor, testosterone as a covariate of interest, and age as a covariate of no interest. These

analyses were conducted in IBM SPSS (v.29; IBM Corp., Armonk, NY, USA).

As mentioned above, we adjusted for possible differences in SNR resulting from the association between testosterone and number of accepted trials by regressing out the square root of the number of trials from each voxel of the whole-brain source images. Separate whole-brain ANCOVAs with sex as a between-subjects factor, testosterone level as a covariate of interest, and age as a covariate of no interest were then performed per oscillatory response using the open-source Statistical Parametric Mapping (SPM 12) toolbox (https://www.fil.ion.ucl.ac.uk/spm/software/) in MATLAB (MathWorks, Natick, MA, USA). A full factorial design was used specifying testosterone interactions with sex. *F*-contrast maps were created individually to analyse for potential main effects of sex, main effects of testosterone, and sex-by-testosterone interaction effects on the underlying neural oscillatory activity serving motor control. To account for multiple comparisons, a two-tailed significance threshold of $P < 0.005$ and cluster ($k$) threshold of 10 contiguous voxels was used to identify significant clusters in all statistical maps (Poline et al., 1995; Worsley et al., 1996, 1999). The $k$-threshold was based on the image smoothness (Barnes et al., 2004) and computed using the fmristat toolbox in MATLAB. In addition, to improve rigor, we also applied the threshold-free cluster enhancement method (TFCE; Smith & Nichols, 2009) to the interaction maps using a cluster-level FWE of $P < 0.05$. Clusters not surviving TFCE are noted below. Follow-up analyses on the interactions were conducted using SPSS. Finally, to reduce the impact of outliers on statistical analyses, participants with values 3 SDs above or below the respective sample mean were excluded.

## Results

### Participant characteristics and behavioural performance

Of the 69 participants recruited for the study, all provided saliva samples. One participant was excluded due to MEG data quality, leaving a total of 68 participants for analysis (mean age = 13.08 years, SD = 1.66 years; 35 female). Across all participants, the mean untransformed testosterone concentration was 37.69 pg/mL (SD = 30.08 pg/mL), and was correlated with age, such that older participants had higher testosterone concentrations ($r_{66} = 0.615$ $P < 0.001$). Since the distribution was non-normal, we log-transformed testosterone (mean = 1.42 pg/mL, SD = 0.40 pg/mL) and this was also correlated with age ($r_{66} = 0.627$, $P < 0.001$; Fig. 2). Males and females did not differ in log-transformed ($t_{66} = 1.81, P = 0.075$) or untransformed ($t_{64} = 1.67$, $P = 0.10$) testosterone, though both were trending higher in males (Table 1).

Overall, participants had a mean accuracy of 93.06% (SD = 6.86%), mean reaction time of 638.35 ms (SD = 203.54 ms), and mean movement duration of 861.11 ms (SD = 175.15 ms; Table 1). Separate $1 \times 2$ ANCOVAs on each behavioural metric with sex as a between-subjects factor, log-transformed testosterone level as a covariate of interest, and age as a covariate of no interest revealed that accuracy was not significantly associated with testosterone ($F_{1,61} = 0.28, P = 0.60$), sex

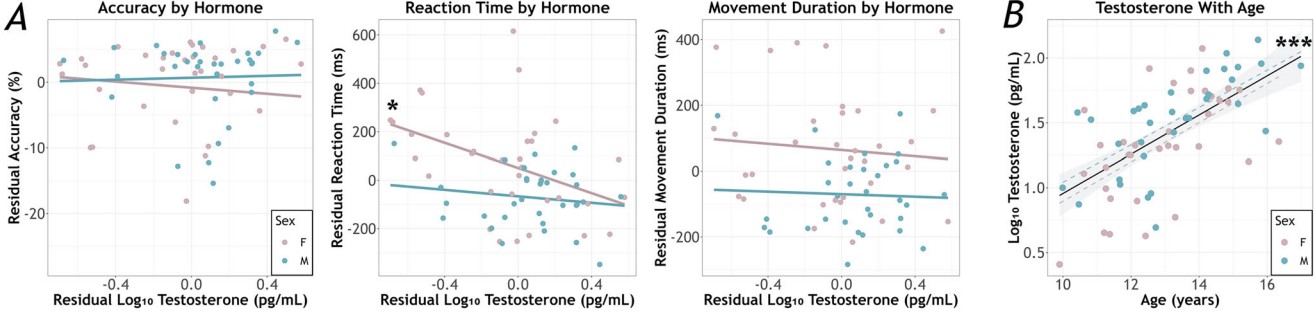

**Figure 2. Behavioural performance and testosterone levels**

*A*, behavioural performance. Percentage correct (accuracy) is shown on the left, reaction time (time between cue to move and movement onset) in the middle, and movement duration (time to complete the sequence) on the right. ANCOVAs revealed that accuracy and movement duration were not associated with sex or testosterone. However, reaction time was significantly faster in males than females, independent of testosterone level. Independent of sex, higher levels of testosterone were associated with faster reaction times. Residuals of behavioural metrics (*y*-axis) and testosterone values (*x*-axis) controlling for age are presented in each plot. Asterisks depict the significance level of the simple slopes, controlling for age. *B*, association between testosterone and age. Older youth had higher testosterone levels. There was no significant difference in testosterone level between males and females (*P* > 0.05). Males (blue) and females (pink) are plotted separately using dashed lines to enhance clarity, and the black trendline represents the entire sample. The grey shaded area represents the standard error of the mean. Asterisks depict the significance level of the simple slope for the combined sample (\**P* < 0.05, \*\*\**P* < 0.001).

**Table 1. Demographic and behavioural data**

| | Full sample | | Females (n = 35) | | Males (n = 33) | | Group statistics | | |
|---|---|---|---|---|---|---|---|---|---|
| | Mean | SD | Mean | SD | Mean | SD | t | F | p |
| Age (years) | 13.08 | 1.66 | 12.92 | 1.58 | 13.25 | 1.75 | 0.82 | — | 0.42 |
| Testosterone (pg/mL) | 37.69 | 30.08 | 31.84 | 26.79 | 43.90 | 32.47 | 1.67 | — | 0.10 |
| Testosterone ($\log_{10}$) | 1.42 | 0.40 | 1.34 | 0.41 | 1.51 | 0.37 | 1.81 | — | 0.075 |
| Accuracy (%) | 93.06 | 6.86 | 92.28 | 7.77 | 93.89 | 5.76 | — | 0.521 | 0.473 |
| Reaction time (ms) | 638.35 | 203.54 | 717.55 | 226.26 | 554.19 | 134.82 | — | 4.516 | 0.038 |
| Movement duration (ms) | 861.11 | 175.15 | 937.38 | 183.19 | 780.06 | 124.24 | — | 1.704 | 0.197 |

($F_{1,61} = 0.52$, $P = 0.47$), or their interaction ($F_{1,61} = 0.26$, $P = 0.62$). There was a main effect of testosterone on reaction time, such that higher levels were associated with faster reaction time ($F_{1,61} = 6.03$, $P = 0.02$), and a main effect of sex ($F_{1,61} = 4.52$, $P = 0.04$), such that males were faster to respond than females, but the sex-by-testosterone interaction was not significant ($F_{1,61} = 2.13$, $P = 0.15$) Finally, movement duration was associated with neither testosterone ($F_{1,61} = 0.22$, $P = 0.64$) nor sex ($F_{1,61} = 1.70$ $P = 0.20$), nor their interaction ($F_{1,61} = 0.17$, $P = 0.69$; Fig. 2).

## MEG sensor-level analyses and source imaging

Sensor-level time–frequency analyses across all participants revealed significant decreases in alpha and beta power (i.e., ERD) relative to baseline from 8–14 Hz and 18–24 Hz, respectively, which began approximately 500 ms prior to movement onset and persisted for about 500 ms after in sensors near the left primary motor cortex (Fig. 3*A*). We also identified a significant increase in power across all participants in the gamma band from 74–84 Hz, which was centred around movement onset from −250 to 250 ms (all $P < 0.0001$, corrected). No other oscillatory responses were significant.

To identify the anatomical regions generating the significant responses identified at the sensor level, we imaged these time–frequency windows using a beamformer. The gamma response was imaged from −250 to 250 ms (74–84 Hz), while alpha (8–14 Hz) and beta (18–24 Hz) were each imaged in two consecutive 500 ms windows (i.e., −500 to 0 ms, 0–500 ms) since the duration of the baseline was only 500 ms (i.e., −2250 to −1750 ms). The output images for alpha and beta were then individually averaged across the two windows per participant, as their spatial distributions were nearly identical by window. Participant-level oscillatory maps were then averaged across all participants, which revealed robust alpha and beta responses in the contralateral primary motor cortex, with a strong gamma response occurring slightly anterior (Fig. 3*B*).

## Whole-brain statistical analyses

To address our primary hypotheses, we conducted whole-brain, voxel-wise ANCOVAs with sex as a between-subjects factor, testosterone level as a covariate of interest, and age as a nuisance covariate. First, there were significant sex-by-testosterone interactions on beta ERD responses in the left anterior prefrontal cortex (PFC; $F_{1,58} = 17.19$, $P < 0.001$) and right dorsomedial PFC ($F_{1,58} = 12.83$, $P < 0.001$). In both cases, the simple slopes controlling for age were significant in males (left PFC: $r_{25} = 0.40$, $P = 0.039$; right PFC: $r_{25} = 0.45$, $P = 0.019$) but not females (left PFC: $r_{31} = -0.25$, $P = 0.160$; right PFC: $r_{31} = -0.23$, $P = 0.194$), with males exhibiting decreases (i.e., less negative) in the strength of beta ERD responses with increasing testosterone. Further, Fisher $r$-to-$Z$ transformations confirmed that these correlation coefficients significantly differed by sex (left PFC: $Z = -2.52$, $P = 0.006$; right PFC: $Z = -2.67$, $P = 0.004$; Fig. 4). There were also significant main effects of sex in both clusters (left PFC: $F_{1,58} = 19.36$, $P < 0.001$; right PFC: $F_{1,58} = 11.86$, $P = 0.001$), but these were superseded by the significant interactions and thus were not further considered. There were no other main effects of sex or testosterone on beta responses. Next, we probed whether beta responses in either of the regions exhibiting interaction effects were significantly correlated with reaction time. When controlling for the effects of age, beta activity in the right dorsomedial PFC was related to task behaviour, with weaker beta responses being associated with faster reaction times in males ($r_{25} = -0.53$, $P = 0.004$) and no relationship in females ($r_{30} = -0.008$, $P = 0.97$). These relationships were significantly different by sex ($Z = 2.15$, $P = 0.016$). Lastly, we conducted exploratory analyses to test whether trial-by-trial variability in either of the significant interaction clusters varied with testosterone and/or sex controlling for age. To this end, we computed the coefficient of variation across trials using the peak voxel time series during the active period in each of the clusters, but the interaction and both main effects were not significant ($P > 0.35$).

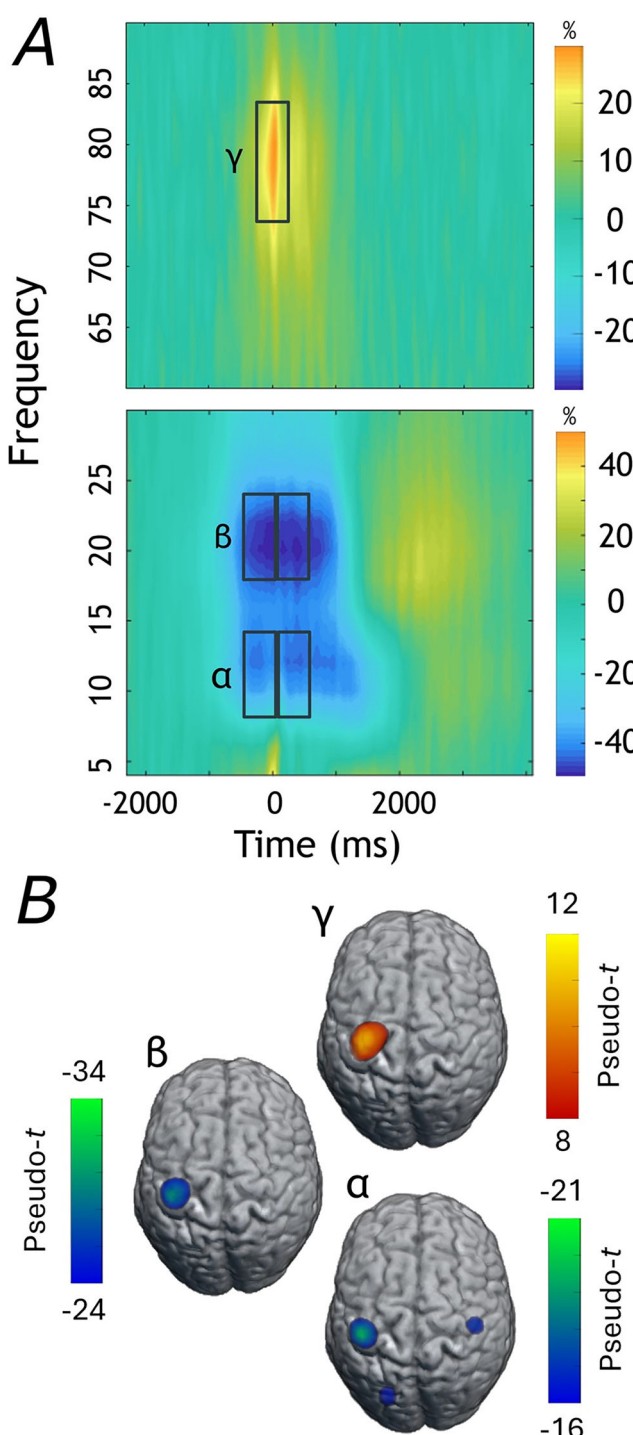

500 ms) and alpha (8–14 Hz, −500 to 500 ms). Additionally, significant increases in gamma power centred on movement onset (i.e., 74–84 Hz, −250 to 250 ms) were observed ($P < 0.0001$, corrected). *B*, grand-averaged beamformer images across all participants. The alpha and beta ERD images were averaged across the two 500 ms windows, as the peak voxel locations were identical between the two windows, with the strongest responses being centred on the contralateral sensorimotor cortices. Similarly, gamma activity was strongest in the contralateral primary motor cortex. Colour scale bars are shown next to each image.

**Figure 3. Time-frequency spectrograms and source reconstruction**

*A*, grand-averaged time–frequency spectrograms across all participants from representative sensors near the left sensorimotor cortex (i.e., MEG0432 for gamma; MEG0443 for alpha and beta). Time in ms is displayed relative to movement onset on the *x*-axis and frequency in Hz on the *y*-axis. Signal power is expressed as a percentage change from baseline. Significant decreases in power relative to the baseline were observed in beta (18–24 Hz, −500 to

The same ANCOVA model on alpha responses also revealed a significant sex-by-testosterone interaction, but the cluster was centred on the right dorsal pre-motor cortex ($F_{1,59} = 10.65$, $P = 0.002$). Consistent with the beta responses, the alpha ERD tended to be weaker with increasing testosterone in males and the opposite in females (Fig. 5), controlling for age. However, evaluation of the simple slopes showed that neither was significant (females: $r_{31} = -0.13$, $P = 0.49$; males: $r_{27} = 0.34$, $P = 0.07$) when controlled for age, although the correlation coefficients did significantly differ from each other ($Z = -1.83$, $P = 0.03$). In addition, similar to the beta analysis, we conducted an exploratory trial-by-trial variability analysis using the peak voxel in the interaction cluster (i.e., right dorsal premotor cortex), but again none of the effects were significant ($P > 0.43$).

There were no main effects of sex on alpha activity, though there were several regions exhibiting significant main effects of testosterone. In the right ventral pre-motor cortex ($F_{1,59} = 10.20$, $P = 0.002$), left inferior parietal ($F_{1,59} = 10.04$, $P = 0.002$), left supramarginal gyrus ($F_{1,59} = 17.07$, $P < 0.001$), and right inferior frontal gyrus ($F_{1,59} = 13.23$, $P < 0.001$), higher testosterone levels were associated with weaker alpha activity across all participants, controlling for age (Fig. 6). Further evaluation of the simple slopes controlling for age showed they were significant in the right ventral pre-motor cortex ($r_{58} = 0.352$, $P = 0.006$, left inferior parietal ($r_{58} = 0.35$, $P = 0.006$, left supramarginal gyrus ($r_{58} = 0.345$, $P = 0.007$, and right inferior frontal gyrus ($r_{58} = 0.354$, $P = 0.006$). Finally, we probed neuro-behavioural correlations at each peak and found that weaker alpha activity in the right ventral premotor cortex was associated with faster reaction times across the full sample ($r_{58} = -0.33$, $P = 0.011$), controlling for age.

Lastly, the whole-brain ANCOVA on gamma activity revealed a significant sex-by-testosterone interaction in the right anterior lobe of the cerebellum, located approximately in lobule V ($F_{1,49} = 9.65$, $P = 0.003$). Note that this cluster did not survive correction when using the TFCE method at $P_{\text{FWE}} < 0.05$. Follow-up evaluation of the simple slopes indicated that the effects of this interaction were such that controlling for the

effects of age, gamma activity in the anterior cerebellum became stronger as testosterone levels increased in males ($r_{21} = 0.53$, $P = 0.010$) but was not related to testosterone in females ($r_{27} = -0.022$, $P = 0.91$). Further, the correlation coefficients differed significantly by sex ($Z = -2.08$, $P = 0.019$; Fig. 7). There was also a significant main effect of sex in this cluster ($F_{1,49} = 13.83$, $P < 0.001$), but this was superseded by the interaction. There were no main effects of testosterone or sex on gamma activity, and the cerebellar peak was not significantly related to behavioural performance. As with the other interaction effects, we followed this up with exploratory analyses of trial-by-trial variability using the peak voxel, but again none of effects were significant ($P > 0.41$).

## Discussion

In this study, we utilized high-density MEG and salivary testosterone to characterize the relationships between pubertal hormone levels and the neural oscillatory dynamics serving performance of motor sequences in typically developing youth. Our rationale included evidence from animal and human studies showing that testosterone affects GABA signalling in the brain (Domonkos et al., 2018; Flores-Ramos et al., 2019), as well as previous studies in developmental cognitive neuroscience showing that testosterone levels are associated with changes in neural oscillatory activity above and beyond age (Fung et al., 2022; Picci et al., 2023).

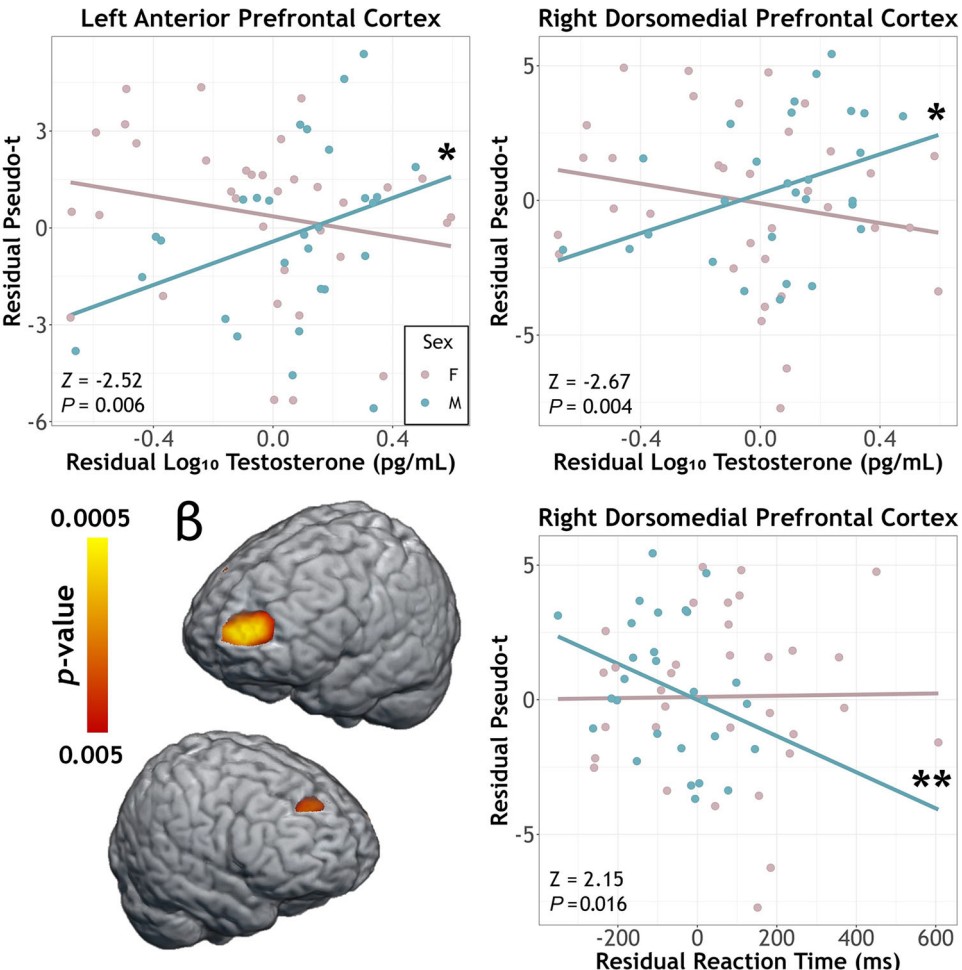

**Figure 4. Sex-by-testosterone interactions on beta ERD responses serving motor control**
A whole-brain ANCOVA revealed significant sex-by-testosterone effects in left and right prefrontal cortices (PFC). In both regions, controlling for age, higher testosterone levels were associated with weaker beta oscillations (i.e., less negative relative to baseline) in males, but were not significantly related to the response in females (top row). Fisher *r*-to-*Z* transformations confirmed the slopes differed in males compared to females in both regions. Further, neurobehavioural correlations revealed a significant association between beta power in the right dorsomedial PFC and reaction time, such that weaker beta oscillations (i.e., less negative) were associated with faster reaction times in males. Scatterplots show residuals of pseudo-*t* on the *y*-axes and log-transformed testosterone or reaction time on the *x*-axes for the sex-by-testosterone effects or neurobehavioural correlation, respectively. Asterisks depict significance level of the simple slopes (*$P < 0.05$, **$P < 0.005$, ***$P < 0.001$).

Consistent with such data, our whole-brain analyses revealed independent effects of testosterone and interactive effects of sex and testosterone on alpha, beta and gamma oscillations in higher-order PFC areas, frontal motor regions and the cerebellum. Follow-up testing revealed significant associations between oscillatory power and task performance in prefrontal and premotor cortices. Below, the implications of these findings are further discussed.

Regarding task performance, reaction time was the only metric associated with testosterone, with faster responses being associated with higher levels. Prior work in our laboratory found that higher levels predicted faster reaction times during single-movement motor tasks (Fung et al., 2022). The main effect of sex in our ANCOVA expands upon this, suggesting that the sex difference is present even beyond the effects of testosterone and age. Interestingly, though both testosterone and sex were associated with reaction time independent of the other, the interaction effect was not significant. This is not entirely surprising, as previous work demonstrated that while puberty was associated with faster reaction times on a visuomotor task, the relationship did not differ by sex (Schulte et al., 2020). Overall, the behavioural results could reflect developmental improvements in some, but not all, aspects of motor performance.

Our most interesting findings were likely the sex-by-testosterone interaction effects on beta activity in bilateral prefrontal cortices, which indicated that higher testosterone was significantly associated with weaker beta ERD in males but not females, when controlling for age. Studies of nonhuman primates point to this region's role in the learning and selection of correct motor sequences (Averbeck et al., 2006; Nakamura et al., 1998; Zhang et al., 2008), and human studies suggest its involvement in working memory, selective attention and response inhibition (Friedman & Robbins, 2022; Szczepanski & Knight, 2014). Regarding the mechanisms, multiple studies have shown that increasing testosterone levels alters GABA signalling (e.g., Domonkos et al., 2018; Flores-Ramos et al., 2019), and previous MEG work using tiagabine to modulate endogenous GABA levels has shown that the motor-related beta ERD response is particularly susceptible to such changes in local GABAergic activity (Muthukumaraswamy et al., 2013). Thus, the current findings may reflect less involvement of these higher-order brain areas in generating motor sequences as testosterone increases in males during adolescence (Heinrichs-Graham et al., 2020). The notion that prefrontal beta becomes weaker with development is also partially supported by prior work (for a review, see Zhang et al., 2025), which found that beta ERD responses weakened with increasing age during a simple motor task in secondary motor regions such as the supplementary motor area (Wilson et al., 2010).

Regarding the sexually dimorphic nature of these effects, previous studies have found sex differences in prefrontal neural activity during inhibitory motor control tasks (Garavan et al., 2006), which were associated with the functional maturation of these regions in a youth sample (Rubia et al., 2013). Other studies have similarly found sex differences in neural activity during a cognitive and motor control task, which were associated with pubertal stage (Schulte et al., 2020). Since females typically begin puberty 1–2 years before males (Goddings et al., 2019), it is possible that we are capturing active cortical

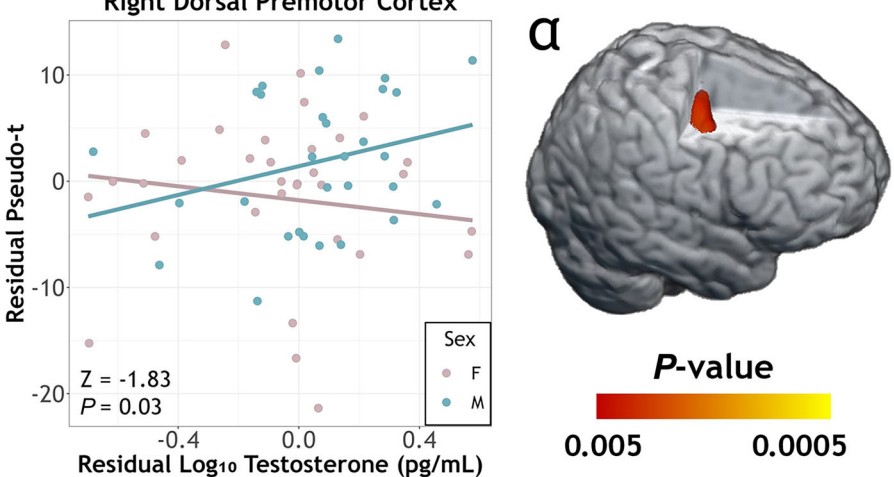

**Figure 5. Sex-by-testosterone interaction on movement-related alpha oscillations**
A whole-brain ANCOVA on alpha oscillatory maps revealed a significant sex-by-testosterone interaction in the right dorsal premotor cortex. With increasing testosterone, males tended to have weaker alpha responses, which was the opposite of that observed in females, although the simple slopes were not significant in either group. The scatterplot axes depict residuals of testosterone (*x*-axis) and pseudo-*t* (*y*-axis), controlling for age.

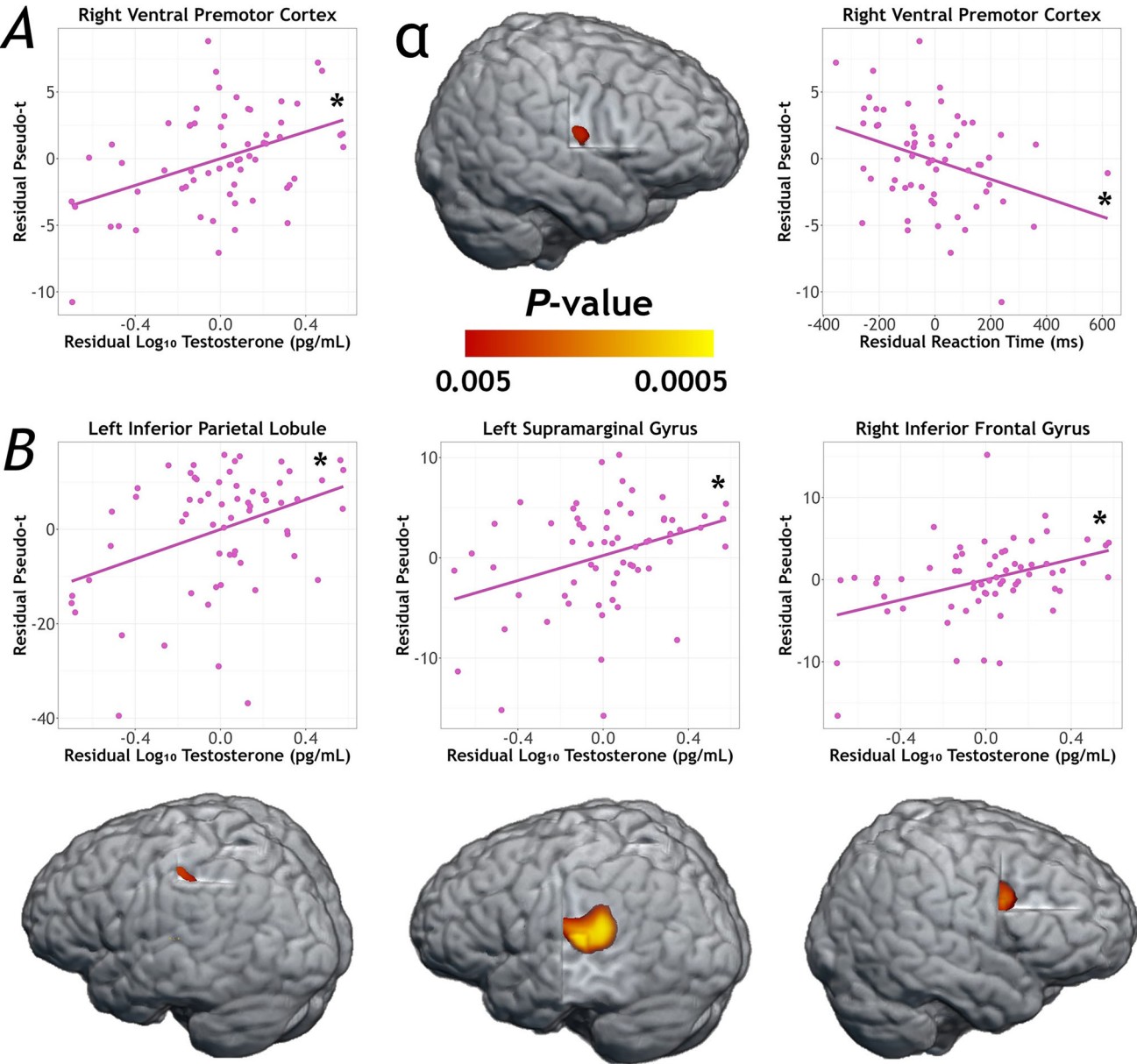

**Figure 6. Main effect of testosterone on movement-related alpha oscillations and neurobehavioural correlations**

*A*, a whole-brain ANCOVA on alpha oscillatory maps revealed a significant main effect of testosterone in the right ventral premotor cortex, such that higher testosterone levels were associated with weaker alpha responses. Partial correlations (right) between alpha activity and reaction time revealed a significant association, such that, controlling for age, weaker alpha responses in the right ventral premotor area were associated with fastest reaction times. Scatterplot axes display the residual values of pseudo-*t* on the *y*-axes, with residuals for testosterone (left plot) and reaction time (right plot) on the *x*-axes, all controlling for the effects of age. *B*, alpha response strength in the left inferior parietal, left supramarginal gyrus, and right inferior frontal gyrus was also associated with testosterone across all participants, such that higher testosterone predicted weaker alpha activity (less negative). Alpha activity was not related to behavioural performance in any of these regions. *x* - and *y*-axes display age-corrected residuals of testosterone level and pseudo-*t*, respectively. Asterisks in all panels depict significance level of simple slopes (*$P < 0.05$, **$P < 0.005$, ***$P < 0.001$).

maturation in males that might have already occurred in females of the same age (Breukelaar et al., 2017; Herting et al., 2015; Koolschijn et al., 2014; Nguyen et al., 2013; Vijayakumar et al., 2014). If so, this could possibly help explain why we observed no relationship in females between beta activity and testosterone level in these higher-order brain regions. Alternatively, these data may suggest that other pubertal hormones are more important in females. For example, progesterone levels are known to increase substantially during the pubertal transition and late adolescent years in females, and like testosterone these changes have been widely connected to alterations in GABA signalling (Delevich et al., 2021; Keating et al., 2019). Further, like developmental changes in testosterone, progesterone changes during the pubertal transition in females have been shown to be particularly relevant for regions characterized by protracted development, including areas of the pre-frontal cortex like those observed here (Gilfarb & Leuner, 2022). Thus, these sex-by-testosterone interaction effects in the PFC could reflect sex differences in the onset and trajectory of testosterone's role in PFC development (i.e., earlier effects in females), sex differences in the critical hormones for PFC development (i.e., testosterone in males and progesterone in females), a combination of these factors, and/or other factors. While additional work is needed to decipher the underlying mechanisms, we propose it is likely a combination of these factors given previous work showing robust changes in spontaneous beta in adolescent females during the resting state (Picci et al., 2023).

Interestingly, in the right dorsomedial prefrontal cortex, we also observed a significant neurobehavioural relationship, which further confirms the importance of this region for task performance. Specifically, we found that weaker beta ERD was significantly associated with faster reaction time when controlling for age in males. This may point to a shift away from the recruitment of prefrontal executive regions as the task becomes less effortful with greater refinement of cognitive abilities during adolescence (Goddings et al., 2019; Wilson et al., 2010). The observation that this neurobehavioural effect was only present in males, with females exhibiting a nearly flat curve, may further support the early maturation of these regions in females, as discussed above, and/or that behavioural changes in females are more closely coupled to other pubertal hormones (e.g., progesterone). In other words, it is possible that relationships with behaviour may have emerged in females had we examined progesterone, and future studies focusing on youth during the pubertal transition should quantify progesterone and testosterone, as well as DHEA if possible. Regarding the latter, previous work in this same age range has shown robust DHEA level by sex interactions in the prefrontal cortices during visual attention tasks (Fung et al., 2022), suggesting DHEA may also play a critical role. Altogether, such work may indicate that the impact of testosterone on performance and motor-related beta oscillations in the PFC occurs earlier in females, with progesterone having a larger effect during puberty and through later adolescence and early adulthood, but further investigations are needed.

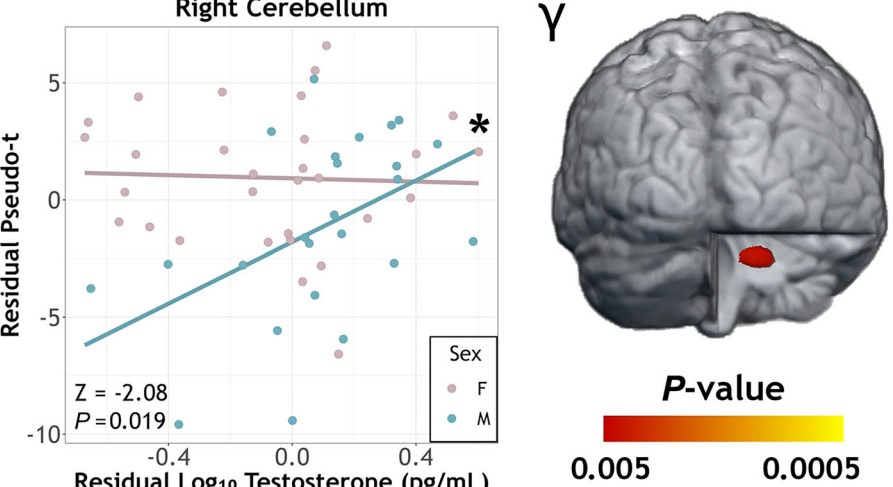

**Figure 7. Sex-by-testosterone interaction on movement-related gamma oscillations**
The results of the ANCOVA model on whole-brain gamma oscillatory maps revealed a significant sex-by-testosterone interaction in the right anterior cerebellum in lobule V. Gamma activity in cerebellar cortices became stronger with increasing testosterone in males, but there was no relationship in females. Fisher *r*-to-*Z* transformation confirmed the slopes differed in males compared to females. The scatterplot axes depict residual values of testosterone (*x*-axis) and pseudo-*t* (*y*-axis), controlling for age. Asterisks display simple slope significance level (*$P < 0.05$, **$P < 0.005$, ***$P < 0.001$).

In the alpha range, we observed a significant sex-by-testosterone interaction in the right dorsal premotor cortex. The dorsal premotor cortex has been proposed to play key roles in motor control, proprioception and motor planning facilitated by sensory integration and action selection (Ben-Shabat et al., 2015; Kantak et al., 2012). Our sex-specific oscillatory findings in the dorsal premotor cortex may be related to its structural maturation, with males displaying thicker cortices at age 9 but a faster rate of cortical thinning than females (Raznahan et al., 2010). We propose that, akin to the prefrontal cortex, sex differences in structural maturation within this region may account for the decrease in alpha power with age observed in males, though without direct observation of testosterone-related functional or structural changes in this region, this interpretation should be taken with caution. Notably, we also found significant main effects of testosterone on alpha oscillations in multiple brain regions, including the left supramarginal gyrus, left inferior parietal, right ventral premotor cortex, and right inferior frontal gyrus. All effects were such that, controlling for age, weaker alpha ERD responses were associated with higher testosterone levels. These brain areas have all been implicated in limb positioning and motor planning (Ben-Shabat et al., 2015; Króliczak et al., 2016), coordination, focusing of attention on visually guided goal-directed movement (Sulpizio et al., 2023; Wang et al., 2014), and fine hand control (Kantak et al., 2012). Further, a significant neurobehavioural association was found in the right ventral premotor cortex between weaker alpha ERD and faster reaction time when controlling for age, which again may suggest less involvement of this higher order motor region with developmental refinement of motor control and the associated circuitry.

Regarding the sex-general nature of our findings, it has been proposed that neural systems develop along a hierarchical sensorimotor-association (S-A) axis (Keller et al., 2023; Luo et al., 2024). In this view, sensorimotor regions would begin to undergo functional refinement before higher-order association cortices. This functional refinement has also been significantly associated with temporally linked structural changes along this S-A axis (Sydnor et al., 2023). This finding corresponds with previous studies, which observed that grey matter begins maturation earlier in somatosensory and motor regions and then later in prefrontal association cortices (Gogtay et al., 2004; Shaw et al., 2008). One possible interpretation of our findings is that these parietal and inferior frontal regions may have already experienced a greater degree of functional and structural refinement as compared to prefrontal regions, which have been shown to begin maturation 1–2 years later (Keller et al., 2023; Larsen & Luna, 2018; Lenroot et al., 2007; Luo et al., 2024; Shaw et al., 2008). These timing differences in the hierarchical refinement of both functional and structural development could in part explain the differing sex relationships observed in our own observations of functional oscillatory development, particularly when considering the age range of our participant sample and the lesser recruitment of these higher order association regions with greater maturation.

We also observed a sex-by-testosterone interaction on gamma activity in the right cerebellum, such that higher levels were significantly correlated with stronger MRGS in males, but not females, when controlling for age. Specifically, we observed this effect in lobule V, which is associated with motor control, sensorimotor integration and somatotopic representation (Kipping et al., 2013; Stoodley & Schmahmann, 2009; Stoodley et al., 2012). In regard to the sex-specificity, previous work has demonstrated significantly earlier maturation of the cerebellum overall in females compared to males (peak volume 11.8 years and 15.6 years respectively), with the anterior lobe in particular exhibiting persistent sex differences into adolescence (Tiemeier et al., 2010). Increased MRGS with greater testosterone levels in males may be related to its particular role in the refinement of GABAergic inhibitory circuitry as a positive allosteric modulator of the GABA-A receptor, which is thought to facilitate gamma oscillatory activity in adolescence (Larsen & Luna, 2018; Luna et al., 2015; Reddy & Jian, 2010; Wang, 2011; Wilson et al., 2007). Additionally, rodent studies suggest that androgen receptors are expressed on cerebellar Purkinje cells (Dart et al., 2024; Qin et al., 2007). A similar distribution of androgen receptors within the human cerebellum could at least partially explain sex-specific effects of testosterone on the oscillatory activity observed in this region. While this could explain the testosterone specific effects, it remains an open question whether similar relationships would have been observed for progesterone in females. As noted above, progesterone changes during the pubertal transition in females are known to affect GABA signalling, which could also modulate the MRGS response in a concentration-dependent manner (Gilfarb & Leuner, 2022). Future work in this area that combines measures from multiple pubertal hormones will be essential to distinguish between these alternatives.

Limitations of the present work are important to acknowledge before closing. First, though we used testosterone concentration as a proxy for development, having measures of pubertal development (e.g., pubertal development scale, PDS) would have been advantageous, since previous studies have demonstrated differences in findings between pubertal stage and endogenous hormone levels (Goddings et al., 2019). Examining both metrics in the same sample would allow better differentiation between broad effects of overall pubertal development and specific effects attributable to

testosterone. Second, our study focused on testosterone, and future studies should quantify DHEA, progesterone and testosterone in the same developmental sample. It is possible that there are sex differences in regard to which hormones show the strongest relationships with neural responses during task performance. Third, testosterone and other steroid hormones exhibit diurnal and menstrual fluctuations in their circulating levels, and saliva and blood-based methods for hormone level quantification may be affected. Future work should consider hair-based measures of steroid hormones, which offer a stable measure of average concentration over a period of 1–3 months, and are not as affected by diurnal or menstrual fluctuations as saliva or blood may be (Fung, Taylor, et al., 2022; Short et al., 2016; Smith et al., 2019; Taylor et al., 2022; Wang et al., 2019). Although care was taken to account for such fluctuations in testosterone levels by collecting at similar times for all participants, hair-based measures would reduce variability and allow for more reliable measurements. Fourth, accuracy on the motor sequencing task was near ceiling (i.e., 93%) and it is possible that effects were missed due to reduced sensitivity. In neurophysiological studies, there is always a trade-off between sensitivity to accuracy effects and participants being accurate enough to ensure adequate epochs remain after excluding incorrect and artifactual trials from the final MEG analyses. In the current case, more difficult sequences could likely have been used without compromising trial counts, and this may have increased sensitivity to accuracy effects. Of note, the most important behavioural variables for this task are generally reaction time (i.e., time to initiate the movement) and movement duration (i.e., time from first movement to completion of the third movement), but accuracy would also be informative, and future studies should consider using more strenuous sequences. Lastly, due to the heterogeneity in the onset and timeline of puberty (Biro et al., 1995, 2014; Goddings et al., 2019), a longitudinal design would allow for better examination of within-subject differences that could relate to pubertal timing and tempo.

In conclusion, our study observed significant relationships between testosterone and oscillatory responses in regions of the prefrontal and cerebellar cortices that have been linked to behavioural improvements in motor control. These improvements may be facilitated by underlying testosterone-related functional and structural refinements that develop in these regions throughout adolescence along a hierarchical pathway, including possible modulation of GABAergic inhibitory signalling. Our study is, to our knowledge, the first to report sex-specific and general effects of testosterone on oscillatory activity throughout motor, somatosensory and attention regions of the brain. These findings emphasize the broad impact of pubertal development on the refinement of neural circuitry during adolescence, above and beyond the effects of chronological age. Further, they strongly support the incorporation and importance of pubertal indices when studying this period of significant motor and neural development.

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

## Additional information

### Data availability statement

The data used in this article have been made publicly available through the Collaborative Informatics and Neuroimaging Suite (COINS; https://coins.trendscenter.org/) framework.

### Competing interests

The authors of this manuscript acknowledge no conflicts of interest, financial or otherwise.

### Author contributions

Conceptualization: E.H.G., T.W. Wilson; Data Collection: G.C.E., D.L.R., A.T.C., E.L.S.; Methodology and Software: T.W. Wilson, E.H.G., N.M.P.; Formal Analysis: J.D., T.W. Ward, N.M.P., T.W. Wilson; Writing - original draft: T.W. Wilson, J.D.,

T.W. Ward; Reviewing and Editing: all authors; Supervision, Resources, and Funding Acquisition: T.W. Wilson, V.D.C., Y.P.W., J.M.S. All authors have read and approved the final version of this manuscript and agree to be accountable for all aspects of the work in ensuring that questions related to the accuracy or integrity of any part of the work are appropriately investigated and resolved. All persons designated as authors qualify for authorship, and all those who qualify for authorship are listed.

## Funding

This research was supported by grants R01-MH121101 (T.W. Wilson), P20-GM144641 (T.W. Wilson), and F30-MH134713 (J.J.S.) from the National Institutes of Health, as well as 1539067 from the National Science Foundation. The funders had no role in study design, data collection, analysis, decision to publish, or preparation of the manuscript.

## Acknowledgements

The funders had no role in study design, data collection, analysis, decision to publish, or manuscript preparation. We want to thank the participants for volunteering to participate in the study and our staff and local collaborators for contributing to the work.

## Keywords

beta oscillations, gamma oscillations, magnetoencephalography (MEG), movement, neurophysiology, puberty

## Supporting information

Additional supporting information can be found online in the Supporting Information section at the end of the HTML view of the article. Supporting information files available:

**Peer Review History**

