## [Peer Review History · The Journal of Physiology]

Testosterone modulates multispectral oscillatory activity serving performance of motor sequences in typically developing youth

Jackson Derby, Thomas W. Ward, Nathan M. Petro, Jake J Son, Grace C. Ende, Danielle L. Rice, Anna T Coutant, Erica Steiner, Vince D. Calhoun, Yu-Ping Wang, Julia Stephen, Elizabeth Heinrichs-Graham, and Tony W. Wilson
DOI: 10.1113/JP289314

Corresponding author(s): Tony Wilson (tony.wilson@boystown.org)

The following individual(s) involved in review of this submission have agreed to reveal their identity: Fumiaki Iwane (Referee #2)

Review Timeline:	Submission Date:	22-May-2025
	Editorial Decision:	04-Aug-2025
	Revision Received:	02-Feb-2026
	Accepted:	24-Feb-2026

Senior Editor: Richard Carson

Reviewing Editor: James Coxon

Transaction Report:

Dear Dr Wilson,

Re: JP-RP-2025-289314 "**Testosterone modulates multispectral oscillatory activity serving complex motor control in typically developing youth**" by Jackson Derby, Thomas W. Ward, Jake J Son, Grace C. Ende, Danielle L. Rice, Anna T Coutant, Erica Steiner, Vince D. Calhoun, Yu-Ping Wang, Julia Stephen, Elizabeth Heinrichs-Graham, and Tony W. Wilson

Thank you for submitting your manuscript to The Journal of Physiology. It has been assessed by a Reviewing Editor and by 2 expert referees and we are pleased to tell you that it is potentially acceptable for publication following satisfactory major revision.

REVISION CHECKLIST:

We look forward to receiving your revised submission.

Yours sincerely,

Richard Carson
Senior Editor
The Journal of Physiology

EDITOR COMMENTS

Reviewing Editor:

Comments to the Author:

Thank you for submitting your work to the Journal of Physiology. Apologies for the delay, I have now received reports from two reviewers with expertise spanning neurodevelopment, neural oscillations, and motor control. The reviewers see some merit in the work but also raise a number of issues that will need to be convincingly addressed to be considered further.

Reviewer 1 sought clarification and transparency in the manuscript as to whether data/participants/equivalent analyses in a slightly different sample size have been reported in your previous publications, in particular Killanin et al., 2023. They also encourage discussion of the findings in relation to animal models linking testosterone to GABAergic inhibition. While there may be some mention of this in the discussion, you might want to consider making this aspect of the discussion more prominent.

Reviewer 2 makes several suggestions for a more thorough treatment of the MEG data. They seek justification for the use of complex demodulation, pointing out that this is less commonly used compared to other approaches. Importantly, they ask for clarification on the use of multiple comparison correction.

For the SPM12 ANCOVA models, it is stated that a cluster threshold of $k = 10$ voxels was used. In agreement with reviewer 2, I would like to see more stringent control for multiple comparisons. This may have implications for the results and discussion. One option is to set "k" such that only significant clusters are retained in the SPM output (set k based on the size of the clusters that are $p < .05$ in the cluster level statistics column of the SPM output. This controls for multiple comparisons at the cluster level (at the chosen significance threshold of $p < .005$).

Please also report the statistics when the result is not significant (e.g. the correlations for females, page 13).

Many of the results demonstrate sex differences. Testosterone is one hormone related to puberty, but there are others (not reported or not measured?) that could impact upon the measured neural oscillations (e.g. progesterone is known to interact with GABA). If other hormones were measured they should also be reported in the manuscript. Some consideration of the impact of female hormones in the discussion seems appropriate, perhaps as an alternative explanation of sex based effects.

Discussion, page 18: "These findings may reflect less involvement of these higher order brain areas in generating complex sequences...". The sequences presented to participants in this study have just 3 components. Consider whether it is appropriate to refer to them as complex.

REFeree COMMENTS

Referee #1:

In this manuscript, Derby and colleagues use a MEG-based paradigm to assess the role of testosterone on neural oscillations related to motor sequencing in typically developing participants aged 10-17 years. The authors report that, when controlling for age, testosterone levels are associated with task-related neural oscillatory activity and showed some associations with task performance (reaction time). The authors report sex-specific effects of testosterone on neural oscillations. In males, higher testosterone was linked to reduced beta activity (beta ERD) in prefrontal cortices and enhanced motor-related gamma oscillations in cerebellar regions. In both sexes, increased testosterone was associated with weaker alpha oscillations in frontal and inferior parietal areas. Several of these neural responses were also significantly related to participant reaction times. Overall, I found this to be a clear and well-written manuscript and the analyses appear sound. Nevertheless, I have some specific comments that I hope the authors can address.

1) Could the authors please clarify whether any data included in the present study has been reported in previous publications from the same lab? I note that Fung et al. (2021, 2022) appear to involve a similar (or possibly overlapping) cohort, albeit using different tasks, while Killanin et al. (2023) appears to employ a similar paradigm (motor sequencing task) and also reports beta activity. If there is any overlap in participants or datasets, it would be important for transparency to clearly state this in the manuscript and to delineate how the current study addresses a distinct scientific question or provides novel insights relative to prior work.

2) Could the authors elaborate on how the present findings relate to previous work such as Killanin et al. (2023), which found that testosterone mediated the effect of age on movement-related beta ERD in the contralateral precentral gyrus? In that study, testosterone mediated the effect of age on both beta ERD and behavioural performance. This finding is noted in the introduction, however, I feel a brief discussion in relation to the current study's findings would enhance interpretability and situate the findings within the existing literature.

Reference:

Killanin et al. (2023). Testosterone levels mediate the dynamics of motor oscillatory coding and behavior in developing youth. *Developmental Cognitive Neuroscience*.

3) Could the authors comment on how the present findings might be interpreted in light of prior evidence from animal models linking testosterone to GABAergic inhibition? I think a brief discussion of whether the observed neural oscillatory patterns could reflect such mechanisms could strengthen the biological plausibility of the findings.

4) The mean task accuracy appears to have been quite high (93.06%). Given that no significant effects were observed for accuracy (only for reaction time), could the authors comment on whether performance may have approached ceiling, potentially limiting sensitivity to group or condition differences in accuracy?

5) Minor: Figure 1: For increased clarity, the authors could consider separating Figure 1A from 1B and 1C, and placing these

latter two figures in the results section rather than the methods as they report empirical findings.

Referee #2:

The manuscript investigates a relationship between testosterone levels and event-related (de)synchronization (ERD/S) during a sequential finger movement task in typically developing youth. The authors have documented their methodologies and presented their results clearly. The discussion thoughtfully contextualizes the findings within existing literature.

While the correlational findings are valuable, the inherent developmental changes in both testosterone levels and ERD/S present an opportunity for future research to explore causal relationships. This point is a significant consideration for advancing our understanding in this field.

Below are specific comments and suggestions for improving the manuscript:

Major Comments:

1. Exploiting Temporal Resolution for Trial-by-Trial Variability: The current analysis relies on averaged ERD/S values per participant. Given the high temporal resolution of MEG data, exploring trial-by-trial fluctuations in ERD/S, and their potential relationship with testosterone, could yield richer insights. It would be valuable to discuss whether the observed relationship holds at a more granular temporal scale, or if averaging obscures important variability.
2. Inclusion of Post-Movement Beta Rebound (PMBR) Analysis: The post-movement beta rebound is a well-established phenomenon associated with neural development and motor control. Expanding the analysis to include PMBR, and investigating its relationship with testosterone, would significantly strengthen the neurodevelopmental implications of the study and provide a more comprehensive picture of motor-related oscillatory activity.
3. Data-Driven Determination of Time Window for ERD/S: The ERD/S characterization employs a fixed 500ms time window. Consideration should be given to determining this window in a data-driven manner, for instance, through cluster-based permutation tests across the time-frequency spectrum. This approach could more accurately capture the peak and duration of the ERD/S effects, potentially revealing more subtle modulations.
4. Data-Driven Determination of Frequency Window: The alpha frequency band is currently fixed at 8-14 Hz. However, Figure 2A suggests weaker modulation in the lower alpha range. It is recommended that the authors consider a data-driven approach to determine the most relevant alpha frequency window, or even subdivide the alpha band (e.g., lower vs. upper alpha) if the data support distinct modulations. This would ensure the analysis is optimally tuned to the observed neural activity.
5. Justification for Complex Demodulation: The authors state the use of complex demodulation, which is less commonly employed for time-frequency decomposition in MEG/EEG research compared to wavelet transforms or short-time Fourier transforms. The manuscript would benefit significantly from a detailed justification for choosing complex demodulation, explicitly outlining its advantages over more commonly used approaches in the context of this specific research question. A brief explanation of the key differences in how these methods handle time-frequency resolution and potential artifacts would also enhance clarity for the readership.
6. Clarification on Multiple Comparison Correction: The methods section currently lacks a detailed explanation of how multiple comparison corrections were performed for the statistical analyses. Given the multispectral and potentially spatio-temporal nature of the data, it is crucial to clearly describe the chosen correction method (e.g., False Discovery Rate (FDR) correction, cluster-based permutation testing, Bonferroni correction) and its application. This information is vital for evaluating the statistical robustness of the reported findings.

END OF COMMENTS

Response to Reviews: JP-RP-2025-289314

EDITOR COMMENTS

Reviewing Editor:

Thank you for submitting your work to the Journal of Physiology. Apologies for the delay, I have now received reports from two reviewers with expertise spanning neurodevelopment, neural oscillations, and motor control. The reviewers see some merit in the work but also raise a number of issues that will need to be convincingly addressed to be considered further.

Thanks for the comment and allowing us to revise the manuscript. We have added substantial new text and recomputed the statistical maps based on your recommendations and the comments of Rev 2, which have broadly strengthened the manuscript. We respond to each comment point-by-point below, noting the line numbers corresponding to changes in the manuscript.

Of note, during the relatively extended review process, both of the co-first authors completed their training and left the laboratory. Thus, we recomputed many of the analyses to fully address the critiques and a person not on the original author list contributed substantially to this. Given this, we are adding a new person (Dr. Nathan Petro) as third author. I corresponded with the editorial staff about this process, and all authors have signed the appropriate form approving this change. We have uploaded the completed form with the submission.

Reviewer 1 sought clarification and transparency in the manuscript as to whether data/participants/equivalent analyses in a slightly different sample size have been reported in your previous publications, in particular Killanin et al., 2023. They also encourage discussion of the findings in relation to animal models linking testosterone to GABAergic inhibition. While there may be some mention of this in the discussion, you might want to consider making this aspect of the discussion more prominent.

As stated in greater detail below in response to Reviewer 1, all data included in this study are from the Developmental Chronnecto-Genomics (Dev-CoG) project (Stephen et al., 2021). Killanin et al. (2023) reported data from 58 participants, and all of these are part of our sample of 68 participants. Importantly, all of these data were completely reanalyzed from the raw data forward and our hypotheses and pipeline were completely different. This is why Dr. Killanin is not a coauthor. Briefly, the Killanin manuscript focused on beta responses in the primary motor cortex and whether testosterone levels mediated the effect of development on these responses. The analysis was completely limited to the beta response in the primary motor cortex and how testosterone affected that. In contrast, the current study followed a data driven approach and imaged two 500 ms alpha windows and two 500 ms beta windows, each centered around the movement onset (i.e., -500 to 0 and 0 to 500 ms), and a high-frequency gamma response time-locked to movement onset. These were then examined at the whole brain level using testosterone as a continuous covariate of interest, sex as a between-subjects factor, and age as a covariate of no interest. Thus, the analysis pipeline completely differed between the two studies, with major differences that include whole-brain analyses (current) versus focus on left primary motor cortex (Killanin), all three motor-related oscillatory responses (alpha, beta, and gamma; current study) versus only the motor beta response (Killanin), and sex as variable of interest (current) versus

sex not being in the model (Killanin). We have added text about these differences between the two studies and the overlapping samples on lines 146-157, 160-169, and 187-191.

Regarding the second question, we appreciate the constructive suggestion and have expanded the discussion to include the link between GABA and testosterone on lines 430-434, 455-459, 477-481, and 546-558.

Stephen JM, Solis I, Janowich J, Stern M, Frenzel MR, Eastman JA, Mills MS, Embury CM, Coolidge NM, Heinrichs-Graham E, Mayer A, Liu J, Wang YP, Wilson TW, Calhoun VD. The Developmental Chronnecto-Genomics (Dev-CoG) study: A multimodal study on the developing brain. *Neuroimage*. 2021 Jan 15;225:117438. doi: 10.1016/j.neuroimage.2020.117438. Epub 2020 Oct 8. PMID: 33039623; PMCID: PMC8045389.

Reviewer 2 makes several suggestions for a more thorough treatment of the MEG data. They seek justification for the use of complex demodulation, pointing out that this is less commonly used compared to other approaches. Importantly, they ask for clarification on the use of multiple comparison correction.

We have responded point-by-point to Reviewer 2 and listed line numbers for each change. First, the Reviewer requested an analysis to probe trial-by-trial variation. This would be very challenging at the whole-brain level, as it would be computationally intensive given the high number of trials, high number of voxels, and the temporal precision. Thus, as a workaround, we conducted such an analysis for the peak voxels in our interaction maps (i.e., the most important findings). We describe the analyses below, but all effects were null. Second, the Reviewer requested a data driven approach to determining the time and frequency windows, which our original paper/analysis already included. Of note, we rounded these to canonical numbers (e.g., 500 ms) because beamforming requires that the time and frequency intervals remain constant across the window and these canonical numbers corresponded closely to the statistical analyses (see Figure 2). Third, the Reviewer recommended that we probe the PMBR response in our imaging analyses. We did not probe the PMBR because it was not significant in our data-driven, sensor-level statistical analysis. As shown in Figure 2, this response was quite weak in our study, potentially due to the response being closely linked with motor termination per trial and there being substantial variation in the movement duration across our developmental sample. Finally, complex demodulation is widely used in the MEG literature, with literally hundreds of publications using this approach. The main motivation for using it over other comparable methods is computational efficiency. Note that this time-frequency transformation only served as the input to our sensor level analyses to derive imaging windows. All of our primary analyses were conducted in voxel space following beamforming, which did not involve complex demodulation. Lastly, we clarify the multiple comparisons correction aspect below.

For the SPM12 ANCOVA models, it is stated that a cluster threshold of $k = 10$ voxels was used. In agreement with reviewer 2, I would like to see more stringent control for multiple comparisons. This may have implications for the results and discussion. One option is to set "k" such that only significant clusters are retained in the SPM output (set k based on the size of the clusters that are $p < .05$ in the cluster level statistics column of the SPM output. This controls for multiple comparisons at the cluster level (at the chosen significance threshold of $p < .005$).

Thanks for the comments and suggestions. Unfortunately, we did not fully understand your suggestion regarding the initial threshold. The initial cluster-forming threshold has been shown

to be perhaps the most critical factor (Eklund et al., 2016), so we assumed you meant that we should use an initial threshold of $p < .005$ (not $p < .05$) and then report the clusters that are significant in the cluster level statistics table at $p < .05$. Of note, there is also a value at the bottom of the table (“FWE_c”) that reports the minimum cluster size for whole-brain error correction of $p < .05$, although this approach appears to be rarely used as we have not seen it reported in previous papers. Since we were not 100% sure about the recommendation, we performed the correction using two different approaches. Before we review these, we want to mention that while our k threshold of 10 voxels may seem small and less rigorous, it is important to remember that our voxel size was 4.0 x 4.0 x 4.0 mm (relatively standard for MEG). Thus, 10 voxels are equivalent to a contiguous activated volume of 640 mm³, which is relatively large and would correspond to a cluster size of 640 voxels in most structural imaging analyses.

We originally used the so-called Worsley method, which is widely implemented in neuroimaging software (e.g., it is the standard method in CONN) and available in the fmristat Matlab toolbox. To estimate, one needs to specify the smoothing kernel, and we used a value of 8 mm. Note that technically beamformer images like those computed here are known to have nonuniform spatial smoothness (Barnes et al., 2004), but virtually all voxels have a FWHM smoothness less than 10 mm (Barnes et al.) and using values between 5-10 mm would not have changed our results in any meaningful way. Using this approach, the k -threshold for a final FWE correction level of $p < .05$ was below 10 voxels for each of our oscillatory maps. Note that our smallest interaction cluster was 17 voxels (i.e., gamma finding in cerebellum) and all others were over 20 voxels, with most being much larger (e.g., see Figure 4, our most important finding). Thus, a much higher k -threshold would have made little difference. Second, we used the threshold-free cluster enhancement method (Smith & Nichols, 2009) with cluster-level FWE set to $p < .05$ and all of our interaction clusters remained, with the exception of the cerebellum which became marginal. Finally, assuming we understood the suggestion above correctly with the initial threshold and cluster column, our results would also hold although the minimum k -threshold changes with each oscillatory map (i.e., each ANCOVA), which we would prefer to avoid since we feel this often confuses readers and leads to misperceptions that one is changing thresholds subjectively based on the comparison. In the manuscript, we have expanded our discussion of the statistical methods and mention that we used TFCE to confirm our interaction clusters, which were our most important findings. See lines 316-324 and 417.

Barnes GR, Hillebrand A, Fawcett IP, Singh KD. Realistic spatial sampling for MEG beamformer images. *Hum Brain Mapp.* 2004 Oct;23(2):120-7. doi: 10.1002/hbm.20047. PMID: 15340934; PMCID: PMC6872013.

Eklund A, Nichols TE, Knutsson H. Cluster failure: Why fMRI inferences for spatial extent have inflated false-positive rates. *Proc Natl Acad Sci U S A.* 2016 Jul 12;113(28):7900-5. doi: 10.1073/pnas.1602413113. Epub 2016 Jun 28. Erratum in: *Proc Natl Acad Sci U S A.* 2016 Aug 16;113(33):E4929. doi: 10.1073/pnas.1612033113. PMID: 27357684; PMCID: PMC4948312.

Smith SM, Nichols TE. Threshold-free cluster enhancement: addressing problems of smoothing, threshold dependence and localisation in cluster inference. *Neuroimage.* 2009 Jan 1;44(1):83-98. doi: 10.1016/j.neuroimage.2008.03.061. Epub 2008 Apr 11. PMID: 18501637.

Please also report the statistics when the result is not significant (e.g. the correlations for females, page 13).

We have updated the results to include such data (line 377).

Many of the results demonstrate sex differences. Testosterone is one hormone related to puberty, but there are others (not reported or not measured?) that could impact upon the measured neural oscillations (e.g. progesterone is known to interact with GABA). If other hormones were measured they should also be reported in the manuscript. Some consideration of the impact of female hormones in the discussion seems appropriate, perhaps as an alternative explanation of sex based effects.

As stated above, these data are from the Dev-CoG project (Stephen et al., 2021). Saliva-based hormone measures were assessed in a subset of participants in this larger study, with testosterone being the primary hormone of interest. Progesterone was never assessed in this study. A smaller sample of the participants had DHEA assessed (see response to first comment of Rev 1 below), but the number with valid DHEA measures in the current sample of 68 participants is less than half and thus we do not feel it would be worthwhile to include these data since any analyses would be underpowered and the DHEA measures were not always based on the same study visit as the testosterone. That said, we agree that consideration of female hormones is important and have included such text in the discussion section on lines 474-487, 495-504, 553-558, 564-567.

Discussion, page 18: "These findings may reflect less involvement of these higher order brain areas in generating complex sequences...". The sequences presented to participants in this study have just 3 components. Consider whether it is appropriate to refer to them as complex.

Thanks for the comment. This is a fair statement and we have adjusted the title of the paper and text throughout to focus on the "sequencing" element and less on the "complex" aspect.

Reviewer #1

In this manuscript, Derby and colleagues use a MEG-based paradigm to assess the role of testosterone on neural oscillations related to motor sequencing in typically developing participants aged 10-17 years. The authors report that, when controlling for age, testosterone levels are associated with task-related neural oscillatory activity and showed some associations with task performance (reaction time). The authors report sex-specific effects of testosterone on neural oscillations. In males, higher testosterone was linked to reduced beta activity (beta ERD) in prefrontal cortices and enhanced motor-related gamma oscillations in cerebellar regions. In both sexes, increased testosterone was associated with weaker alpha oscillations in frontal and inferior parietal areas. Several of these neural responses were also significantly related to participant reaction times. Overall, I found this to be a clear and well-written manuscript and the analyses appear sound. Nevertheless, I have some specific comments that I hope the authors can address.

We thank Reviewer 1 for the constructive comments. We address each point-by-point below.

1) Could the authors please clarify whether any data included in the present study has been reported in previous publications from the same lab? I note that Fung et al. (2021, 2022) appear to involve a similar (or possibly overlapping) cohort, albeit using different tasks, while Killanin et al. (2023) appears to employ a similar paradigm (motor sequencing task) and also reports beta

activity. If there is any overlap in participants or datasets, it would be important for transparency to clearly state this in the manuscript and to delineate how the current study addresses a distinct scientific question or provides novel insights relative to prior work.

We are sorry this was not clearer in the original draft. We originally stated these data were derived from the Developmental Chronnecto-Genomics (Dev-CoG) project (Stephen et al., 2021), but did not mention the individual studies that have used these data. Regarding Fung et al., the 2021 manuscript in *Developmental Cognitive Neuroscience* and the 2022 manuscript in *Neuroimage* included all 68 participants in the current sample plus additional participants. Importantly, these Fung et al. papers were based on a more basic visual gratings task and their additional participants did not complete the motor sequencing task reported in the current study; thus, they are not included in our sample. Of note, Fung et al. also had a 2022 paper in *Human Brain Mapping* that used a visuospatial attention task. This work focused on how DHEA levels affected visual attention and included 35 participants, with 29 of these being in the current study. Finally, Killanin et al. (2023) reported data from 58 participants, and all of these are part of our sample of 68 participants. Importantly, all of these data were completely reanalyzed from the raw data forward using a different analysis pipeline to pursue distinct hypotheses. This is why Dr. Killanin is not a coauthor on the current study. Briefly, the Killanin et al. (2023) focused on beta responses in the primary motor cortex and whether testosterone levels mediated the effect of development on these responses. The MEG analysis was completely limited to the beta response in the peak voxel within the left primary motor cortex. In contrast, the current study followed a data driven approach and imaged two 500 ms alpha windows and two 500 ms beta windows, each centered around the movement onset (i.e., -500 to 0 and 0 to 500 ms), and a high-frequency gamma response that coincided with movement onset. These were then examined at the whole brain level using a voxel-wise ANCOVA with testosterone as a continuous covariate of interest, sex as a between-subjects factor, and age as a covariate of no interest. Thus, the analysis pipeline completely differed between the two studies, with key differences that included whole-brain analyses (current) versus focus on a single voxel within the left primary motor cortex (Killanin), three motor-related oscillatory responses (alpha, beta, and gamma; current study) versus peri-movement beta response (Killanin), and sex as variable of interest (current) versus sex not being in the model (Killanin). We have added text about these differences between the two studies on lines 146-157 and 160-169 of the introduction and lines 187-191 of the methods.

2) Could the authors elaborate on how the present findings relate to previous work such as Killanin et al. (2023), which found that testosterone mediated the effect of age on movement-related beta ERD in the contralateral precentral gyrus? In that study, testosterone mediated the effect of age on both beta ERD and behavioural performance. This finding is noted in the introduction, however, I feel a brief discussion in relation to the current study's findings would enhance interpretability and situate the findings within the existing literature. Reference: Killanin et al. (2023). Testosterone levels mediate the dynamics of motor oscillatory coding and behavior in developing youth. *Developmental Cognitive Neuroscience*.

We agree that expanding on the findings of Killanin et al. (2023) in the introduction would help frame the current study and thank the reviewer for their constructive comments. We have added such discussion on lines 150-157 and 161-169.

3) Could the authors comment on how the present findings might be interpreted in light of prior evidence from animal models linking testosterone to GABAergic inhibition? I think a brief discussion of whether the observed neural oscillatory patterns could reflect such mechanisms could strengthen the biological plausibility of the findings.

Thanks for the suggestion. We have added such to the revised Discussion section on lines 430-432, 455-459, 475-478, and 546-558. Of note, much of the data linking testosterone and GABAergic activity in humans has been in the area of sex differences in anxiety and depressive disorders, so we have navigated this carefully to ensure we don't overstep the data in regard to extensions to motor control.

4) The mean task accuracy appears to have been quite high (93.06%). Given that no significant effects were observed for accuracy (only for reaction time), could the authors comment on whether performance may have approached ceiling, potentially limiting sensitivity to group or condition differences in accuracy?

We agree that accuracy was near ceiling. In neurophysiological studies, there is always a tradeoff between sensitivity to accuracy effects and participants being accurate enough to ensure there are enough epochs remaining after excluding incorrect trials. In other words, it is standard to exclude trials where participants responded inaccurately from the final MEG analyses and if participants are incorrect on too many trials, then there are not enough epochs remaining to compute reliable beamformer source images. On the other hand, if participants are correct too often then the study lacks sensitivity to look at accuracy effects, which is likely the case here. We have added this as a potential limitation of the study on lines 575-584. Of note, the most important behavioral variables for this task are reaction time (i.e., time to initiate the movement) and movement duration (i.e., total time to complete all three movements), and reaction time was related to testosterone level and sex, but not their interaction. Duration was not related to either variable.

5) Minor: Figure 1: For increased clarity, the authors could consider separating Figure 1A from 1B and 1C, and placing these latter two figures in the results section rather than the methods as they report empirical findings.

We agree this would improve clarity. We have separated the figures accordingly in the revised manuscript. Note this resulted in all other figure numbers changing.

Again, we thank Reviewer #1 for their time and constructive comments on our manuscript, which have significantly improved the final product.

Reviewer #2

The manuscript investigates a relationship between testosterone levels and event-related (de)synchronization (ERD/S) during a sequential finger movement task in typically developing youth. The authors have documented their methodologies and presented their results clearly. The discussion thoughtfully contextualizes the findings within existing literature.

While the correlational findings are valuable, the inherent developmental changes in both testosterone levels and ERD/S present an opportunity for future research to explore causal relationships. This point is a significant consideration for advancing our understanding in this field.

Below are specific comments and suggestions for improving the manuscript:

We thank Reviewer 2 for this time and constructive review of our experimental work. We respond to each comment point-by-point below.

Major Comments:

1. Exploiting Temporal Resolution for Trial-by-Trial Variability: The current analysis relies on averaged ERD/S values per participant. Given the high temporal resolution of MEG data, exploring trial-by-trial fluctuations in ERD/S, and their potential relationship with testosterone, could yield richer insights. It would be valuable to discuss whether the observed relationship holds at a more granular temporal scale, or if averaging obscures important variability.

We agree that trial-by-trial analyses are often insightful and have conducted such analyses in past studies (e.g., Spooner & Wilson, 2022). However, such analyses are substantially more complex in the current framework as our approach considers whole-brain, voxel-wise effects related to testosterone and sex, whereas past studies looking at trial-by-trial effects have usually either remained at the MEG sensor level (i.e., scalp level) or focused on a region-of-interest or peak voxel. For example, Spooner & Wilson (2022) focused on time series from the peak voxel in the primary motor cortex. Thus, one approach to test trial-by-trial variability effects in the current context is to follow-up some of our key findings in the whole-brain analyses by extracting voxel time series and evaluating trial level data. To this end, we extracted peak voxel time series for each of the testosterone-by-sex interaction peaks (i.e., beta peaks in the left anterior prefrontal and right dorsomedial prefrontal cortex, alpha in the right dorsal medial premotor cortex, and gamma in the cerebellum) and computed the ERD/S amplitude trial-by-trial for each region. These data were then used to compute the trial-wise coefficient of variation (CoV) per region and participant. These CoV values were then examined using the same ANCOVA approach as the initial whole-brain models with testosterone and sex as factors of interest and controlling for age. Unfortunately, neither the interaction nor either main effect was significant in any of the regions. This was somewhat surprising to us, so we conducted exploratory analyses as a sanity check to test whether the CoV was simply correlated with age, which indicated that the CoV for both beta peaks were indeed slightly correlated with age, while no such relationship was seen for the alpha or gamma peak. In sum, these analyses suggest that trial-by-trial variability in these regions varies with age, but that this is not closely tied to testosterone level or sex. In the revised manuscript, we mention the null findings for the trial-by-trial variability analyses for the interaction peaks on lines 388-392, 399-401, and 424-426. We do not mention the age-based correlations as these are outside the scope of our study, which was focused on the effects of testosterone and sex above and beyond age.

2. Inclusion of Post-Movement Beta Rebound (PMBR) Analysis: The post-movement beta rebound is a well-established phenomenon associated with neural development and motor

control. Expanding the analysis to include PMBR, and investigating its relationship with testosterone, would significantly strengthen the neurodevelopmental implications of the study and provide a more comprehensive picture of motor-related oscillatory activity.

We fully agree that the PMBR is a well-established phenomenon. In fact, we have published many papers describing the PMBR (e.g., Heinrichs-Graham et al., 2017), including how it is affected by age (Trevarrow et al., 2019) but not testosterone level (Fung et al., 2022) during development. Importantly, our MEG sensor level analyses included the whole epoch and thus the time period of the PMBR was included. In fact, the PMBR can be discerned in Figure 3A (formerly Figure 2A) as the ERS that is viewable in the beta range from 2000-2500 ms. We did not further examine this response through beamformer imaging (etc.) because our analysis pipeline was data driven and the response was not significantly different from baseline following permutation testing. Thus, technically, we were not justified to image the response in our data-driven framework since it was not significant at the sensor level. Regarding why this common response may not have been significant in the current study, we would speculate that this was due to the task, as the multistep motor sequence was associated with considerable variation in movement duration. Essentially, the PMBR is known to be tightly coupled to motor termination (Heinrichs-Graham et al., 2017; many other papers) and in the current study this varies with movement duration. Thus, the wide range of durations may have smeared the response over several hundred milliseconds ultimately reducing its amplitude. While this is just speculation, it is supported by the prolonged duration of the beta ERS in Figure 3A (i.e., 1700-2800+ ms).

Fung MH, Heinrichs-Graham E, Taylor BK, Frenzel MR, Eastman JA, Wang YP, Calhoun VD, Stephen JM, Wilson TW. The development of sensorimotor cortical oscillations is mediated by pubertal testosterone. *Neuroimage*. 2022 Dec 1;264:119745. doi: 10.1016/j.neuroimage.2022.119745. Epub 2022 Nov 9. PMID: PMC9816764.

Heinrichs-Graham E, Kurz MJ, Gehringer JE, Wilson TW. The functional role of post-movement beta oscillations in motor termination. *Brain Struct Funct*. 2017 Sep;222(7):3075-3086. doi: 10.1007/s00429-017-1387-1. Epub 2017 Mar 24. PMID: PMC5610915.

Trevarrow MP, Kurz MJ, McDermott TJ, Wiesman AI, Mills MS, Wang YP, Calhoun VD, Stephen JM, Wilson TW. The developmental trajectory of sensorimotor cortical oscillations. *Neuroimage*. 2019 Jan 1;184:455-461. doi: 10.1016/j.neuroimage.2018.09.018. Epub 2018 Sep 12. PMID: PMC6230487.

3. Data-Driven Determination of Time Window for ERD/S: The ERD/S characterization employs a fixed 500ms time window. Consideration should be given to determining this window in a data-driven manner, for instance, through cluster-based permutation tests across the time-frequency spectrum. This approach could more accurately capture the peak and duration of the ERD/S effects, potentially revealing more subtle modulations.

We thank Reviewer 2 for this comment. As alluded to in the previous comment, we did determine the time-frequency windows using a data-driven approach (lines 474-482) in the original analysis. As stated there, the specific time-frequency bins used for source reconstruction were determined using a mass univariate approach based on the general linear model. To reduce the risk of false-positive results while maintaining reasonable sensitivity, a two-stage procedure was followed to control for Type-1 error. In the first stage, two-tailed paired-sample t-tests against baseline were conducted on each data point, and the output spectrogram of t-values was thresholded at $p < 0.05$ to define time-frequency bins containing potentially significant oscillatory deviations across all participants. In stage two, time-frequency bins that survived the

threshold were clustered with temporally and/or spectrally neighboring bins that were also above the threshold ($p < 0.05$), and a cluster value was derived by summing the t-values of all data points in the cluster. Nonparametric permutation testing was then used to derive a distribution of cluster values, and the significance level of the observed clusters (from stage one) were tested directly using this distribution (Ernst, 2004; Maris & Oostenveld, 2007). This testing was conducted from 4-90 Hz across all participants to assess for theta, alpha, beta, and high gamma frequency bands, which have been previously linked to motor processes (Fung, Heinrichs-Graham, et al., 2022; Wilson et al., 2010). For each comparison, 10,000 permutations were computed. Based on these analyses, the time-frequency windows containing significant oscillatory events relative to baseline across all participants were beamformed. For further details on our data processing pipeline, see (Wiesman & Wilson, 2020).

Regarding why we used round numbers like -500 to 0 ms, this reflects several constraints. First, the resolution of our time-frequency transform was 25 ms in the temporal domain and 2.0 Hz in the frequency domain. Thus, our statistical analysis was done on pixels that were 25 ms wide. Second, our baseline was defined as a 500 ms duration window from -2250 to -1750 ms; this window was chosen to maximize the distance from the motor offset of the previous trial and the motor onset of the next trial to ensure it was not contaminated by lingering PMBR activity (previous trial) or the emergence of the beta ERD response (upcoming trial). The beamforming approach that we used computes noise normalized images and requires a baseline or passive window of equal duration and bandwidth to the active or target window (text is marked on lines 294-296). Thus, the maximum duration window that we could image was 500 ms. Since both the alpha and beta extended over 500 ms, we divided these into two windows and imaged them separately (see lines 363-365). Finally, beamformer imaging requires that the time and frequency window remains constant across the imaging window. Thus, selecting the precise window for imaging requires one to determine the core time window and frequency band from the statistical results. In other words, the significant cluster following permutation testing varies in bandwidth across the significant time period and the investigator must determine the “core” time-frequency band for imaging. In the current study we took the approach of requiring that the “core” bands only include significant pixels, which required that we not include the edges that varied across the window. For example, the beta response in the 18-24 Hz band was significant from -500 to 500 ms, but within this 1000 ms window there were intermittent time periods where the 16-18 Hz pixel was also significant and these were not included. Further, we did not go beyond 500 ms because the bandwidth of the beta response started to systematically narrow at about 500 ms. Lastly, since the significant bands did approximate the relatively round numbers of -500 and +500 for both alpha and beta (see Figure 3), we chose these to streamline and simplify the presentation. That said, shifting the window by 25 or 50 ms would have made virtually no difference in our results, as we ultimately averaged the output images from -500 to 0 ms with those from 0 to 500 ms (per band) since the images were virtually identical (text is marked on lines 363-365; also stated in Figure 3 caption). While we focused on alpha and beta in this section, the same principles applied to the gamma window selected for imaging.

4. Data-Driven Determination of Frequency Window: The alpha frequency band is currently fixed at 8-14 Hz. However, Figure 2A suggests weaker modulation in the lower alpha range. It is recommended that the authors consider a data-driven approach to determine the most relevant alpha frequency window, or even subdivide the alpha band (e.g., lower vs. upper alpha) if the

data support distinct modulations. This would ensure the analysis is optimally tuned to the observed neural activity.

As noted in the previous response, we used a data driven approach. The selected alpha window (i.e., 8-14 Hz) was significant from -500 to 500 ms. While the response appears weaker from 8-10 Hz in Figure 3A, it was still significant and was stronger in other sensors. We focused on this sensor in Figure 3 because it captured both the alpha and beta responses relatively well. Further, while this sensor suggests that the alpha duration extended beyond 500 ms, this was not generally the case and the significant passband varied substantially after 500 ms. Thus, we focused on the core alpha response due to the reasons noted in the previous response.

5. Justification for Complex Demodulation: The authors state the use of complex demodulation, which is less commonly employed for time-frequency decomposition in MEG/EEG research compared to wavelet transforms or short-time Fourier transforms. The manuscript would benefit significantly from a detailed justification for choosing complex demodulation, explicitly outlining its advantages over more commonly used approaches in the context of this specific research question. A brief explanation of the key differences in how these methods handle time-frequency resolution and potential artifacts would also enhance clarity for the readership.

We appreciate the question and admit we did not give a detailed description of complex demodulation (Kovach and Gander, 2016; Papp and Ktonas, 1977). Briefly, complex demodulation works by first transforming the signal into the frequency space, using a Fast Fourier Transform (FFT). This results in a frequency spectrum, inherently containing the same power and cross spectrum information as the original signal. From here, this frequency spectrum is (de)modulated in a step-wise manner to adopt the center frequency of a series of complex sinusoids with increasing carrier frequencies, in a process termed "heterodyning." These resulting signals are then low-pass filtered to reduce spectral leakage, and thus the nature of this filter inherently determines the time and frequency resolution of the resulting data. For this study, the time-frequency analysis was performed with a frequency-step of 2 Hz and a time-step of 25 ms between 4 and 90 Hz, using a 4 Hz lowpass finite impulse response (FIR) filter. We have added a more detailed description of complex demodulation on lines 257-268.

Regarding why we chose to use complex demodulation, we would first point out that while not as common as FFT or wavelets, complex demodulation has been used in hundreds (perhaps thousands) of MEG and EEG papers. It is quite common and often preferred when using high density arrays because it is very computationally efficient. Other major advantages include low spectral leakage and better transient detection. For nonstationary signals (like MEG and EEG), complex demodulation is often better for extracting features such as specific frequency bands, arrival times, and the envelope of underlying oscillations in noisy data (Kovach & Gander).

Kovach CK, Gander PE. The demodulated band transform. *J Neurosci Methods*. 2016 Mar 1;261:135-54. doi: 10.1016/j.jneumeth.2015.12.004. Epub 2015 Dec 19. PMID: 26711370; PMCID: PMC5084918.

Papp N, Ktonas P. Critical evaluation of complex demodulation techniques for the quantification of bioelectrical activity. *Biomed Sci Instrum*. 1977 Apr 25-27;13:135-45. PMID: 871500.

6. Clarification on Multiple Comparison Correction: The methods section currently lacks a detailed explanation of how multiple comparison corrections were performed for the statistical

analyses. Given the multispectral and potentially spatio-temporal nature of the data, it is crucial to clearly describe the chosen correction method (e.g., False Discovery Rate (FDR) correction, cluster-based permutation testing, Bonferroni correction) and its application. This information is vital for evaluating the statistical robustness of the reported findings.

We are sorry this was not clearer in the original draft. First, as stated above in response to question #3, our sensor level analyses used a nonparametric permutation testing approach to control for multiple comparisons. The goals of these analyses were to identify the time-frequency windows for beamforming using a data-driven approach. Once these time-frequency windows (i.e., oscillatory responses) had been imaged, we examined our primary hypotheses using whole-brain, voxel-wise ANCOVAs with sex as a between subjects factor, testosterone level as a covariate of interest and age as a nuisance covariate. We thresholded the resulting maps at $p < .005$ and controlled for multiple comparisons using a cluster criterion (i.e., k -threshold) of 10 voxels (i.e., 640 mm^3). For clarity, using a cluster approach controls the rate of false positives across the whole-brain volume based on the theory of Gaussian fields (Worsley et al., 1996, 1999; Poline et al., 1995). Briefly, with an uncorrected threshold of .005, one would expect five false positives per thousand comparisons. Since our whole-brain maps had $\sim 10,000$ voxels, that would translate to 50 randomly distributed false positives across the brain. Thus, requiring that significant clusters reach the $p < .005$ threshold and be comprised of at least ten contiguous voxels (i.e., voxels that share a side or corner) strongly increases the statistical rigor of the resulting analyses by leveraging the random distribution of false positives. Regarding the basis of the k -threshold (i.e., 10 voxels), this is based on the image smoothness. Given comments from the editor, we recomputed the k -threshold using the `fmristat` toolbox in Matlab, which confirmed our cluster size was adequate. Of note, in the current study, our significant clusters were relatively large across all models and so the particular choice of k -threshold had virtually no effect. Finally, given the concerns of the editor and this reviewer, we also reassessed all of our interaction clusters using the threshold-free cluster enhancement method (Smith & Nichols, 2009), which further supported our conclusions. We have added text to the methods on lines 474-482 (sensor level analyses) and 316-324 (source level analyses), as well as the results on line 417 to improve clarity on all of these elements.

To close, we thank Reviewer #2 for their suggestions and constructive comments. We hope our responses fully addressed the concerns and feel the changes and additions to the manuscript resulted in significant improvements.

Poline JB, Worsley KJ, Holmes AP, Frackowiak RS, Friston KJ. (1995). Estimating smoothness in statistical parametric maps: Variability of p values. *Journal of Computer Assisted Tomography*, 19(5), 788–796. <https://doi.org/10.1097/00004728-199509000-00017>

Smith SM, Nichols TE. Threshold-free cluster enhancement: addressing problems of smoothing, threshold dependence and localisation in cluster inference. *Neuroimage*. 2009 Jan 1;44(1):83-98. doi: 10.1016/j.neuroimage.2008.03.061. Epub 2008 Apr 11. PMID: 18501637.

Worsley KJ, Andermann M, Koulis T, MacDonald D, Evans AC. (1999). Detecting changes in nonisotropic images. *Human Brain Mapping*, 8(2–3), 98–101. [https://doi.org/10.1002/\(SICI\)1097-0193\(1999\)8:2/3<98::AID-HBM5>3.0.CO;2-F](https://doi.org/10.1002/(SICI)1097-0193(1999)8:2/3<98::AID-HBM5>3.0.CO;2-F)

Worsley KJ, Marrett S, Neelin P, Vandal AC, Friston KJ, Evans AC. (1996). A unified statistical approach for determining significant signals in images of cerebral activation. *Human Brain Mapping*, 4(1), 58–73. [https://doi.org/10.1002/\(SICI\)1097-0193\(1996\)4:1<58::AID-HBM4>3.0.CO;2-O](https://doi.org/10.1002/(SICI)1097-0193(1996)4:1<58::AID-HBM4>3.0.CO;2-O)

Dear Dr Wilson,

Re: JP-RP-2026-289314R1 "**Testosterone modulates multispectral oscillatory activity serving performance of motor sequences in typically developing youth**" by Jackson Derby, Thomas W. Ward, Nathan M. Petro, Jake J Son, Grace C. Ende, Danielle L. Rice, Anna T Coutant, Erica Steiner, Vince D. Calhoun, Yu-Ping Wang, Julia Stephen, Elizabeth Heinrichs-Graham, and Tony W. Wilson

We are pleased to tell you that your paper has been accepted for publication in The Journal of Physiology.

Yours sincerely,

Richard Carson
Senior Editor
The Journal of Physiology

IMPORTANT POINTS TO NOTE FOLLOWING ACCEPTANCE OF YOUR PAPER:

- **IMPORTANT NOTICE ABOUT OPEN ACCESS:** To assist authors whose funding agencies mandate immediate public access to published research findings, The Journal of Physiology allows authors to pay an Open Access (OA) fee to have their papers made freely available immediately on publication.

The Corresponding Author will receive an email from Wiley with details on how to register or log in to Wiley Authors where you will be able to place an order.

- You can check if your funder or institution has a Wiley Open Access Account here:
<https://authors.wiley.com/author-resources/Journal-Authors/open-access/author-compliance-tool.html>

- You can help your research get the attention it deserves! Check out Wiley's free Promotion Guide for best-practice recommendations for promoting your work at: www.wileyauthors.com/eoo/guide. You can learn more about Wiley Editing Services which offers professional video, design, and writing services to create shareable video abstracts, infographics, conference posters, lay summaries, and research news stories for your research at: www.wileyauthors.com/eoo/promotion.

- If you would like to receive our 'Research Roundup', a monthly newsletter highlighting the cutting-edge research published in The Physiological Society's family of journals (The Journal of Physiology, Experimental Physiology, Physiological Reports, The Journal of Nutritional Physiology and The Journal of Precision Medicine: Health and Disease), please click this link, fill in your name and email address and select 'Research Roundup':
<https://www.physoc.org/journals-and-media/membernews>

EDITOR COMMENTS

Reviewing Editor:

The authors have made satisfactory edits to the manuscript.

REFeree COMMENTS

Referee #1:

The authors have now addressed all of my previous comments. I have no further queries.

Referee #2:

The authors addressed my previous comments. I have no further comments.